# Adversarial Perturbations Are Formed by Iteratively Learning Linear Combinations of the Right Singular Vectors of the Adversarial Jacobian

**Thomas Paniagua** [1]  **Chinmay Savadikar** [1]  **Tianfu Wu** [1]

Code:  https://github.com/ivmcl/ordered-topk-attack

## Abstract

White-box targeted adversarial attacks reveal core vulnerabilities in Deep Neural Networks (DNNs), yet two key challenges persist: (i) How many target classes can be attacked simultaneously in a specified order, known as the *ordered top-K attack* problem ($K \geq 1$)? (ii) How to compute the corresponding adversarial perturbations for a given benign image directly in the image space? We address both by showing that *ordered top-K perturbations can be learned via iteratively optimizing linear combinations of the right singular vectors of the adversarial Jacobian* (i.e., the logit-to-image Jacobian constrained by target ranking). These vectors span an orthogonal, informative subspace in the image domain. We introduce **RisingAttacK**, a novel Sequential Quadratic Programming (SQP)-based method that exploits this structure. We propose a holistic figure-of-merits (FoM) metric combining attack success rates (ASRs) and $\ell_p$-norms ($p = 1, 2, \infty$). Extensive experiments on ImageNet-1k across six ordered top-$K$ levels ($K = 1, 5, 10, 15, 20, 25, 30$) and four models (ResNet-50, DenseNet-121, ViT-B, DEiT-B) show RisingAttacK consistently surpasses the state-of-the-art QuadAttacK.

## 1. Introduction

Deep Neural Networks (DNNs) have witnessed tremendous progress across numerous applications, enabling the recent development of large foundation models (such as DeepMind's AlphaZero and AlphaFold and OpenAI's ChatGPT) that are widely recognized to pave a promising way towards Artificial General Intelligence (AGI). Despite of the remarkable achievement, adversarial vulnerability (Szegedy et al., 2013; Goodfellow et al., 2014) remains *the Achilles heel of all DNNs*, particularly in computer vision, as revealed by white-box adversarial attacks, especially targeted white-box attacks (Carlini & Wagner, 2017) that can fool trained DNNs towards arbitrarily specified targets. With the access to network architectures and pretrained weights, white-box attacks can expose their deep vulnerabilities and test their robustness. In practice, white-box attacks are also used as surrogate models in learning transferrable black-box (Inkawhich et al., 2019; Li et al., 2020a; Naseer et al., 2021; Zhao et al., 2023; Fang et al., 2024) and no-box (Li et al., 2020b) attacks. So, seeking more powerful white-box attacks will provide a foundation both for learning potentially stronger black-box and no-box attacks. In this paper, we focus on learning white-box targeted attacks in ImageNet-1k (Russakovsky et al., 2015) classification tasks.

We consider the generalized setting of targeted attacks, **ordered top-K attacks** (Zhang & Wu, 2020; Paniagua et al., 2023), that relax the traditional top-1 targets (e.g., to fool a DNN to classify a dog image as a cat) to $K$ targets ($K \geq 1$) in any given orders (e.g., to fool a DNN to classify a dog image with [car, tree, table] as the ordered top-3 prediction, see the middle in Fig. 1). Ordered top-K targeted attacks expose deeper vulnerabilities of DNNs, since they show the manipulability of the decision boundary of DNNs at the logits subspace levels, especially when $K$ is large (e.g., $K > 20$). These attacks are particularly impactful in applications where the order of predictions significantly influences outcomes, such as recommendation systems or multi-class decision-making, and adversaries can exploit decision hierarchies to disrupt critical processes. Particularly, safety-critical systems (e.g., face unlock, medical triage, content moderation) reason over *entire ranked lists*. An attacker dictating *all* top predictions (similar in spirit to [cat, car, fish] vs only "cat") obtains finer control and evades simple "Top-1 changed" detectors.

In the meanwhile, **security evaluations now recommend $K > 1$.** For example, in the new differential-privacy evaluation guideline, NIST SP 800-226 (March 2025) (Near

[1]Department of Electrical and Computer Engineering, North Carolina State University, Raleigh, USA. Correspondence to: Thomas Paniagua <tapaniag@ncsu.edu>, Tianfu Wu <twu19@ncsu.edu>.

*Proceedings of the 42$^{nd}$ International Conference on Machine Learning*, Vancouver, Canada. PMLR 267, 2025. Copyright 2025 by the author(s).

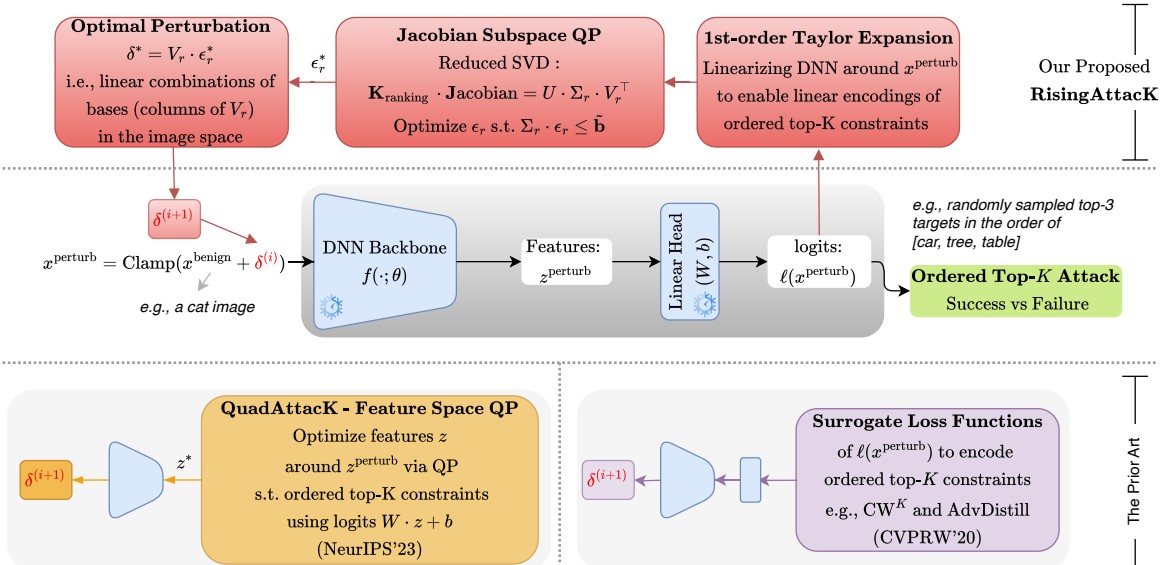

**Figure 1.** Workflow comparisons between our proposed image space based RisingAttacK (top) and the prior art (bottom), $\text{CW}^K$ and AdvDistill (Zhang & Wu, 2020) and QuadAttacK (Paniagua et al., 2023) for learning ordered top-K targeted adversarial attacks (Zhang & Wu, 2020) for a benign image $x^{\text{benign}} \in [0, 1]^D$ (e.g., $D = 3 \times 224 \times 224$). See text for details.

et al., 2025) devotes an entire discussion to "Practical differentially-private Top-K selection" and cites (Durfee & Rogers, 2019) as its canonical example which repeatedly frames robustness/utility checks around whether the *entire ordered set* of the highest-scoring items is preserved under noise—not just the single best—underscoring regulators' need for *Top-K mis-ranking tests*. **Ordered top-K attacks thus supply the stress-test regulators and practitioners request but that Top-1-only methods cannot deliver.**

Ordered top-K attacks can be straightforwardly formulated as an optimization problem with highly non-linear constraints, which is intractable in the vanilla form (see Eqn. 4). Thus, learning ordered top-K attacks poses a unique challenge as they require the perturbations to precisely influence the model's ranking mechanism across multiple outputs ($K > 1$), not just a single decision ($K = 1$). Addressing ordered top-K attacks offers valuable insight into how models distribute their confidence across multiple classes and the vulnerabilities associated with this ranking structure. To address this challenge, there are two main approaches in the prior art (see the bottom of Fig. 1):

- **Designing surrogate loss functions**, such as the $\text{CW}^K$ (extended from the CW method (Carlini & Wagner, 2017)) and the Adversarial Distillation method proposed in (Zhang & Wu, 2020), that transform the constrained optimization problem to an unconstrained one.
- **Reformulating the non-linear constraints to linear ones**, such as the recently proposed QuadAttacK (Paniagua et al., 2023), by first solving the optimization problem in the feature space of the DNN backbone (i.e., the input space to the linear head classifier), and then back-

propagating the optimized features through the backbone to compute adversarial perturbations.

QuadAttacK has shown significant improvement in comparison with methods based on surrogate loss functions. While QuadAttacK is effective, its effectiveness diminishes significantly when $K > 20$ and the computing budget is restricted (e.g., 30 steps). It relies on backpropagation to map the optimized feature space perturbation back to the original input image space. This introduces an indirect connection between the optimization problem and the resulting image space perturbation, leading to limitations as-follows:

- **Feature vs. Image Space Misalignment:** Minimizing the perturbation in the feature space does not always correspond to minimizing it in the image space due to the nonlinear mapping between the two spaces.
- **Suboptimal Visual Perturbations:** The resulting adversarial examples may not fully align with the visual characteristics of the image, as perturbations that minimize the distance in feature space may not correspond to minimal or visually coherent changes in the image space, due to the nonlinear relationship between the two spaces.

To the best of our knowledge, no existing approaches have been proposed for learning ordered top-K attacks ($K \geq 1$) directly in the image space due to the complexities of high-dimensional, non-linear optimization. Potentially due to this, it remains unresolved to seek an explicit formula for "seeing" what adversarial perturbations are formed, if possible. In this paper, **we propose a Sequential Quadratic Programming (SQP) formulation to address the non-linear optimization challenge of learning ordered top-$K$**

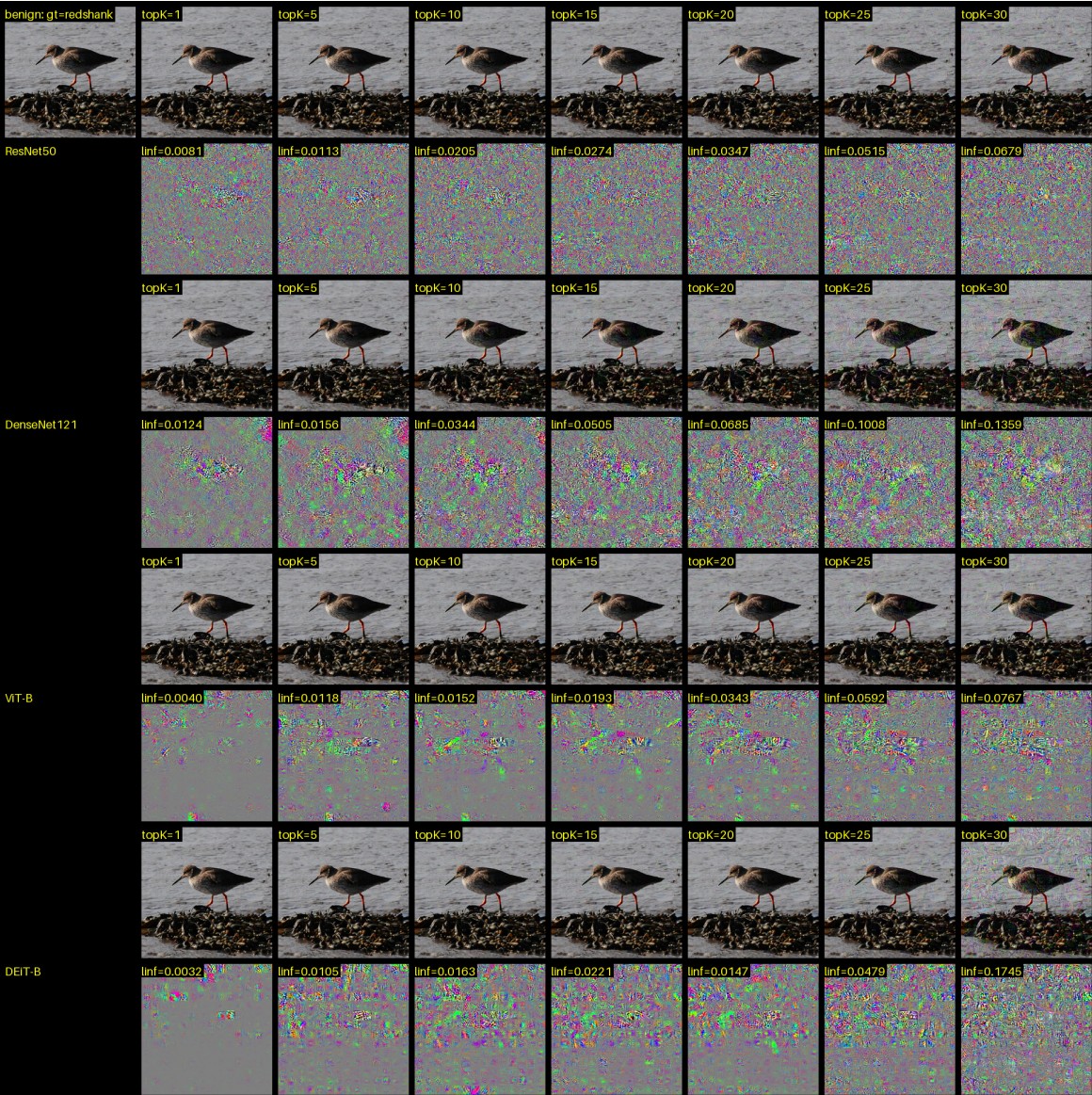

*Figure 2.* Examples of adversarial examples and associated perturbations learned for a benign image (ILSVRC2012_val_00002633 with the ground-truth label, `redshank`) by our RisingAttacK using a list of randomly sampled 30 targets in the order of: `mask`, `analog-clock`, `slide-rule`, `Siberian-husky`, `harmonica`, `African-chameleon`, `dowitcher`, `hyena`, `wing`, `pillow`, `garter-snake`, `Great-Pyrenees`, `puffer`, `banana`, `West-Highland-white-terrier`, `whippet`, `brown-bear`, `snowplow`, `tarantula`, `space-heater`, `sports-car`, `jean`, `sandbar`, `perfume`, `papillon`, `triceratops`, `barrow`, `peacock`, `digital-watch`, `carton`. The adversarial perturbations are normalized to $[0, 1]$ for the sake of visualization. Some of them are treated as being "visually imperceptible" based on the commonly used threshold $8/255 = 0.0314$ for $\ell_\infty$ ('linf') norms. For the benign image, the top-30 predictions by the four models respectively are:

• **ResNet50**: redshank, ruddy turnstone, red-backed sandpiper, dowitcher, oystercatcher, grey whale, red-breasted merganser, crane, sea lion, chainlink fence, lakeside, wreck, quail, partridge, screwdriver, plastic bag, pelican, parachute, killer whale, sulphur-crested cockatoo, African crocodile, white stork, pole, bucket, caldron, hummingbird, sandbar, king penguin, nail, syringe.

• **DenseNet121**: redshank, ruddy turnstone, red-backed sandpiper, oystercatcher, breakwater, dowitcher, sea lion, academic gown, abaya, mortarboard, red-breasted merganser, lifeboat, cloak, espresso, lipstick, theater curtain, wood rabbit, umbrella, refrigerator, ruffed grouse, king penguin, partridge, sandbar, diamondback, hen-of-the-woods, wine bottle, mailbox, stone wall, volcano, redbone.

• **ViT-B**: redshank, red-backed sandpiper, ruddy turnstone, dowitcher, oystercatcher, water ouzel, Madagascar cat, chain saw, apiary, red-breasted merganser, Tibetan mastiff, cicada, seat belt, American egret, wall clock, mask, snow leopard, schipperke, potter's wheel, lycaenid, mud turtle, curly-coated retriever, dumbbell, television, strainer, feather boa, buckle, junco, boa constrictor, volcano.

• **DEiT-B**: redshank, ruddy turnstone, red-backed sandpiper, dowitcher, oystercatcher, red-breasted merganser, warthog, worm fence, Indian elephant, African crocodile, maze, badger, snowplow, American black bear, stone wall, king penguin, car wheel, rock python, water ouzel, guillotine, wild boar, centipede, diamondback, apiary, barrow, horned viper, sundial, guenon, bustard, skunk.

**attacks directly in the image space**, as illustrated in Fig. 1 (top), which can address the drawbacks of QuadAttacK (Paniagua et al., 2023). Our approach efficiently solves the SQP problem by iteratively computing the singular value decomposition (SVD) of the adversarial Jacobian (i.e., the attack-targets-ranking constrained logit-to-image Jacobian matrix), obtained from linearizing the DNN during optimization. This direct optimization in image space provides deeper insights into the learned adversarial perturbations: **ordered top-$K$ adversarial perturbations can be learned by iteratively optimizing linear combinations of the right singular vectors (corresponding to non-zero singular val- ues) of the adversarial Jacobian**. The proposed method is thus dubbed as `RisingAttacK` (see examples in Fig. 2). Our proposed RisingAttacK achieves significant better performance than the prior state-of-the-art method, QuadAttacK (Paniagua et al., 2023) in experiments.

## 2. Related Work and Our Contributions

**Adversarial Attacks.** Adversarial attacks aim to expose the vulnerabilities of DNNs by introducing small, often visually imperceptible perturbations to input data that cause the model to produce incorrect or adversary-specified outputs. Foundational work in adversarial machine learning introduced methods for generating adversarial examples under various norms and constraints, including the Fast Gradient Sign Method (FGSM) (Goodfellow et al., 2014), Projected Gradient Descent (PGD) (Madry et al., 2017), and the Carlini-Wagner (CW) attack (Carlini & Wagner, 2017). These early approaches primarily targeted top-1 classification outputs, seeking to force the model to misclassify an input into a specific target class.

Beyond top-1 attacks, researchers have investigated adversarial perturbations that manipulate the top-$K$ predictions of a model. (Zhang & Wu, 2020) introduced one of the earliest methods for addressing **ordered** top-$K$ adversarial attacks, focusing on creating an optimal target class distribution aided by word embedding vectors, and minimizes KL divergence to this optimal distribution that satisfies the ordered top-$K$ objective. (Tursynbek et al., 2022) explored the geometry of **unordered** top-$K$ adversarial attacks, highlighting the complexities of crafting perturbations that adhere to top-$K$ constraints. (Reza et al., 2025) proposed GSBA$^K$, a geometric score-based unordered top-$K$ blackbox attack method built on (Reza et al., 2023). (Paniagua et al., 2023) advanced this area by formulating the ordered top-$K$ adversarial attack problem as a quadratic programming (QP) optimization in the feature space. This approach efficiently enforced the desired ordering of logits but required back-propagation to map feature space solutions to the image space. Our proposed JacAttacK builds upon these foundations by extending the idea in QuadAttacK (Pani-

agua et al., 2023) to directly address the ordered top-$K$ adversarial attack problem in the image space.

**Sequential Quadratic Programming (SQP).** SQP is a widely used framework for solving nonlinear constrained optimization problems (Nocedal & Wright, 1999). By iteratively solving QP subproblems that linearize constraints and use a quadratic approximation of the objective, SQP effectively handles problems involving nonlinearities and complex constraint sets (Boggs & Tolle, 2000). This approach is particularly relevant in high-dimensional settings, such as adversarial attacks, where the constraints often involve intricate relationships between model outputs. However, applying SQP to large-scale problems, such as those in image space, can be computationally expensive due to the need to repeatedly compute gradients and solve large QPs (Gill et al., 2005). Our method adapts SQP for adversarial optimization by leveraging subspace splitting to reduce the dimensionality of the optimization problem, thereby overcoming scalability challenges while preserving accuracy.

**Our Contributions.** The main contributions of this paper are as-follows: (i) **Novel Theoretical Insights:** It introduces explicit derivations connecting adversarial perturbations to singular vectors of the adversarial Jacobian, providing new theoretical clarity. (ii) **Methodological Innovation:** It is the first method to directly optimize ordered top-K adversarial attacks in image space via SQP, significantly improving alignment between optimized solutions and visually coherent perturbations. (iii) **Empirical Advances:** It provides comprehensive evaluation across multiple architectures and attack levels, consistently outperforming the previous state-of-the-art, QuadAttacK using a proposed holistic metric, Figure of Merits (FoM) covering both success rates and perturbation magnitudes.

## 3. Approach

In this section, we first define the problem of learning ordered top-K attacks (Zhang & Wu, 2020), and then present details of our proposed RisingAttacK.

### 3.1. Problem Definition

**Model Under Attack.** Let $(x^{\text{benign}}, y) \in [0, 1]^{3 \times H \times W} \times \mathcal{Y}$ be a pair of a benign RGB image $x^{\text{benign}}$ with spatial height and width, $H$ and $W$ respectively, and its ground-truth label $y$ with the $C$-class label space $\mathcal{Y} = \{1, \cdots, C\}$. Let $D = 3 \times H \times W$ be the dimension of the input image space. In ImageNet-1k (Russakovsky et al., 2015) classification, we have $C = 1000$ and $D = 3 \times 224 \times 224 \approx 1.5e5$.

A DNN trained for image classification is a highly-nonlinear mapping from the image space to the logit space:
$$\ell(\cdot; \Theta) : [0, 1]^D \to \mathbb{R}^C, \tag{1}$$
where $\Theta$ collects all learned parameters of the DNN. We

will omit $\Theta$ in notations and use $\ell(\cdot)$ for simplicity.

We consider validation or testing images that can be correctly classified by a trained DNN such as ResNet-50 (He et al., 2016) in learning attacks, i.e., $y = \arg\max \ell(x^{\text{benign}})$. The DNN is frozen in learning attacks.

**The Adversarial Region of Ordered Top-K Targeted Attacks for $(x^{\text{benign}}, y)$.** Let $\mathcal{T} \in \mathcal{Y} \setminus \{y\}$ be a randomly sampled sequence of ordered top-$K$ targets for attacking $x^{\text{benign}}$, $K = |\mathcal{T}|$. The adversarial region is defined by,

$$\mathcal{R}(x^{\text{benign}}, \mathcal{T}) = \Big\{ x^{\text{adv}} \in [0, 1]^D; \text{ satisfying}$$

$$\ell(x^{\text{adv}})_{t_i} > \ell(x^{\text{adv}})_{t_{i+1}}, t_i \in \mathcal{T}, i \in [1, K-1], \quad (2)$$

$$\ell(x^{\text{adv}})_{t_K} > \ell(x^{\text{adv}})_j, t_K \in \mathcal{T}, \forall j \in \mathcal{Y} \setminus \mathcal{T} \Big\}, \quad (3)$$

where the subscript represent the entry index of the logit vector. *We often expect the perturbation energy, defined by $l_p$-norm, $||x^{adv} - x^{benign}||_p$, is as small as possible to be visually imperceptible for $p = 1, 2, \infty$.* An adversarial perturbation $\delta = x^{\text{adv}} - x^{\text{benign}}$ is treated as being "visually imperceptible" based on the commonly used threshold $\ell_\infty < 8/255 = 0.0314$.

**Encoding Ordered Top-$K$ Targeted Attack Constraints in the Logit Space.** Denote by $\mathbb{K} \in \{+1, 0, -1\}^{(C-1) \times C}$ the matrix that encodes ordered top-$K$ constraints subject to $\mathcal{T}$, with which the adversarial region can be rewritten by, $\mathcal{R}(x^{\text{benign}}, \mathcal{T}) = \{x^{\text{adv}} \in [0,1]^D; \text{ satisfying } \mathbb{K} \cdot \ell(x^{\text{adv}}) > 0\}$.

**Learning ordered top-$K$ attacks** for a benign image $x^{\text{benign}}$ can be posed as a constrained minimization problem,

$$\minimize_{\delta \in \mathbb{R}^D} \quad ||\delta||_p, \quad (4)$$

$$\text{subject to} \quad \mathbb{K} \cdot \ell(x^{\text{perturb}}) > 0,$$

$$x^{\text{perturb}} = \text{Clamp}(x^{\text{benign}} + \delta),$$

where $\delta$ is the adversarial perturbation variables, $|| \cdot ||_p$ represents the $l_p$-norm (typically, $l_2$-norm is used). Clamp$(\cdot)$ ensures the perturbed example $x^{\text{perturb}}$ is in the input image space (i.e., $x^{\text{perturb}} \in [0, 1]^D$) via element-wise pixel value clipping. **The challenge of solving Eqn. 4 lies in the nonlinear constraints caused by the highly non-linear DNN (Eqn. 1).** *In practice, we also expect the learning of $x^{adv}(= x^{benign} + \delta^*) \in \mathcal{R}(x^{benign}, \mathcal{T})$ is efficient subject to a predefined and limited budget such as 30 or 60 iterations.*

### 3.2. Our Proposed RisingAttacK

Inspired by the QP approach in QuadAttacK (Paniagua et al., 2023) (see a brief overview in Appendix A), but different from its feature space QP formulation, we aim to solve Eqn. 4 directly in the image space under the SQP framework (Boggs & Tolle, 2000). The core idea is to iteratively linearize the nonlinear constraints in Eqn. 4. Due to the large number of constraints, $C - 1$ and the high dimension-

ality of the image space, $D$, which make the optimization with constraints linearized still infeasible in practice, we streamline yet retain the solutions of Eqn. 4.

Eqn. 4 can be re-expressed as,

$$\minimize_{x \in [0,1]^D} \quad ||x - x^{\text{benign}}||_p, \quad (5)$$

$$\text{subject to} \quad \mathbb{K} \cdot \ell(x) > 0,$$

Similar in spirit to QuadAttacK (Paniagua et al., 2023) and all other attack methods, our proposed RisingAttacK is an iterative optimization algorithm starting from the initial perturbed image $x^{\text{perturb}} = \text{Clamp}(x^{\text{benign}} + \delta^{(0)})$ (e.g., $\delta^{(0)} = 0$). At the $i$-th iteration, let $x^{\text{perturb}} = \text{Clamp}(x^{\text{benign}} + \delta^{(i)})$ be the current perturbed image. We omit the iteration index in $x^{\text{perturb}}$ for simplicity. To solve Eqn. 5, our RisingAttacK is streamlined as follows:

- We linearize the DNN $\ell(\cdot)$ around the current perturbed image $x^{\text{perturb}}$, so the nonlinear constraints $\mathbb{K} \cdot \ell(x) > 0$ become linear. We use the first-order Taylor expansion,

$$\ell(x) \approx \ell(x^{\text{perturb}}) + \mathbb{J}(x^{\text{perturb}}) \cdot (x - x^{\text{perturb}}), \quad (6)$$

where $\mathbb{J}(x^{\text{perturb}}) \in \mathbb{R}^{C \times D}$ is the **logit-to-image Jacobian matrix** of the DNN, **which represents the sensitivity of the DNN logits with respect to changes in $x^{\text{perturb}}$. Each row of $\mathbb{J}(x^{\text{perturb}})$ corresponds to the gradient of a particular logit with respect to the input pixels.**

- After the linearization, there is a gap between the objective function (i.e., $x$ should be as close as possible to the benign image), and the linearized constraints which entails $x$ to be sufficiently close to the perturbed image $x^{\text{perturb}}$ to ensure the linearization is sufficiently approximately accurate to retain the ordered top-K constraints. We re-express the objective function $||x - x^{\text{benign}}||$ to be $||x - x^{\text{anchor}}||$, where $x^{\text{anchor}}$ represents the anchor in optimization, $x^{\text{anchor}} = x^{\text{benign}}$ or $x^{\text{anchor}} = x^{\text{perturb}}$ (the current perturbed image). We propose an anchor selection strategy: we start with $x^{\text{anchor}} = x^{\text{perturb}}$ so the algorithm can quickly reach the adversarial region, that is to find $x \in \mathcal{R}(x^{\text{benign}}, \mathcal{T})$. We then seek better adversarial images with smaller perturbation energies by letting the anchor $x^{\text{anchor}} = x^{\text{benign}}$. The two steps may iterate based on monitoring the improvement with respect to a threshold (see Sec. 3.2.4). Consider $l_2$-norm for the objective, Eqn. 5 is re-expressed as,

$$\minimize_{x \in [0,1]^D} \quad ||x - x^{\text{anchor}}||_2^2, \quad (7)$$

$$\text{s.t.} \quad \mathbb{K} \cdot \big(\ell(x^{\text{anchor}}) + \mathbb{J}(x^{\text{anchor}}) \cdot (x - x^{\text{anchor}})\big) > 0.$$

- Eqn. 7 is theoretically solvable, but not practically feasible since the number of constraints, $C - 1$ is large (e.g., $C = 1000$) and the dimension of variables, $D$ is extremely high (e.g., $D = 3 \times 224 \times 224$), especially given the limited budgets in learning attacks. We propose methods to address these challenges.

### 3.2.1. COMPACT ORDERED TOP-K CONSTRAINTS

We introduce a mapping that condenses the logit space without compromising the ordered top-$K$ constraints, but allows the number of rows of the Jacobian matrix to only depend on the nubmer of targets, $K$.

To that end, we first notice that Eqn. 3 can be simplified to reduce the number of constraints from $C - K$ to 1 without breaking the overall ordered top-$K$ constraints,

$$\ell(x)_{t_K} > \max\big(\{\ell(x)_j\}_{j \in \mathcal{Y} \setminus \mathcal{T}}\big), \qquad (8)$$

where $\max(\cdot)$ introduces nonlinearity in the constraints with a gradient switching effect in learning that is not desirable, however. We tackle this by introducing a mapping,

$$G : \ell(\cdot) \in \mathbb{R}^C \to \mathbf{l}(\cdot) \in \mathbb{R}^{d=K+M+1}, \qquad (9)$$

where $M$ is a multiplicative of $K$ such as $M = 5 \cdot K$. The mapping $G$ reorders the logits and augments them with a differentiable nonlinear term (see Appendix B for details due to space limit).

Denote by $\mathbf{K} \in \{+1, 0, -1\}^{(d-1) \times d}$ the compact encoding matrix using the mapping $G$, which has a nice form with rows rotating from $\begin{bmatrix} 1 & -1 & 0 & \cdots & 0 \end{bmatrix}$ (i.e., the logits in $\mathbf{l}(\cdot)$ are expected to decreasingly ordered). $\mathbf{K}$ remains unchanged in the optimization.

With the mapping $G$ reordered and condensed logits $\mathbf{l}(\cdot)$, Eqn. 6 is redefined by,

$$\mathbf{l}(x) \approx \mathbf{l}(x^{\text{anchor}}) + \mathbf{J}(x^{\text{anchor}}) \cdot (x - x^{\text{anchor}}), \qquad (10)$$

where the Jacobian matrix $\mathbf{J}(x^{\text{anchor}}) \in \mathbb{R}^{d \times D}$ with $d = K + M + 1$ only dependent on the number of attack targets, $K$, and often $d \ll C$ (e.g., $d = 101$ for $K = 20$ with $C = 1000$ in ImageNet-1k).

### 3.2.2. JACOBIAN SUBSPACE QP

With the compact encoding matrix $\mathbf{K}$ and the updated Taylor expansion (Eqn. 10), the constraints in Eqn. 7 are then simplified and we have,

$$\underset{x \in [0,1]^D}{\text{minimize}} \quad ||x - x^{\text{anchor}}||_2^2, \qquad (11)$$

$$\text{subject to} \quad A \cdot x \leq \mathbf{b},$$

where $A = -\mathbf{K} \cdot \mathbf{J}(x^{\text{anchor}})$ incorporates the ordered top-K ranking constraints into the logit-to-image sensitivity analysis (i.e., **the adversarial Jacobian**), $\mathbf{b} = \mathbf{K} \cdot \big(\mathbf{l}(x^{\text{anchor}}) - \mathbf{J}(x^{\text{anchor}}) \cdot x^{\text{anchor}}\big) + \mathbf{m}$ defines the constraint boundaries and the feasibility of the optimization, with $\mathbf{m}$ being margins introduced to control the target separability and to change from strict '$<$' to '$\leq$' in optimization constraints. Here, $A \in \mathbb{R}^{(d-1) \times D}$, $\mathbf{b} \in \mathbb{R}^{d-1}$. $\{x \in \mathbb{R}^D; A \cdot x \leq \mathbf{b}\}$ defines a high-dimensional **polyhedron** in the image space.

Directly solving Eqn. 11 is still computationally challenging and does not meet the low budget in learning attacks. We exploit the structure of the polyhedron via projection.

**Exploiting the Subspace Structure of $A$.** We utilize the structure of $A$ revealed by its SVD,

$$A = U \cdot \Sigma \cdot V^\top = U \cdot \Sigma_r \cdot V_r^\top, \qquad (12)$$

where $U \in \mathbb{R}^{(d-1) \times (d-1)}$, $\Sigma \in \mathbb{R}^{(d-1) \times D}$, and $V \in \mathbb{R}^{D \times D}$. $\Sigma = \begin{bmatrix} \Sigma_r & \mathbf{0} \\ \mathbf{0} & \mathbf{0} \end{bmatrix}$ is a diagonal matrix with singular values, $\text{diag}(\sigma_1, \cdots, \sigma_{d-1})$. $U$ (and $V$) provide orthogonal bases for the column (and the row) spaces of $A$. And, $U \cdot U^\top = \mathbb{I}$ and $V^\top \cdot V = \mathbb{I}$ (where $\mathbb{I}$ represents the identity matrix). The rows of $U$ corresponding to large singular values identify the most sensitive ranking constraints. The row space of $A$ corresponds to the input image space, and **each column of $V$ represents a principle direction in the image space**. Since we have $d \ll D$, we can drop the last $D - (d-1)$ columns of $V$ to form the *reduced* SVD, i.e., $V_r \in \mathbb{R}^{D \times (d-1)}$, the first $d - 1$ columns of $V$, which consists of the $d - 1$ orthogonal bases in the image space, and spans the entire solution space of the polyhedron defined by $A \cdot x \leq \mathbf{b}$. **The columns of $V_r$ span a subspace in which adversarial perturbations are most effective towards satisfying ordered top-$K$ constraints.** Learning ordered top-K attacks can be achieved in the subspace accordingly, as we solve it in the following.

Let $\delta = x - x^{\text{anchor}}$, Eqn. 11 is rewritten as,

$$\underset{\delta \in \mathbb{R}^D}{\text{minimize}} \quad ||\delta||_2^2, \qquad (13)$$

$$\text{subject to} \quad A \cdot \delta \leq \mathbf{b} - A \cdot x^{\text{anchor}},$$

With the change of variables $\delta = V \cdot \epsilon$, we have,

$$||\delta||_2 = ||V \cdot \epsilon||_2 = ||\epsilon||_2, \text{ (since $V$ is orthogonal)} \qquad (14)$$

$$A \cdot \delta = U \cdot \Sigma \cdot V^\top \cdot V \cdot \epsilon = U \cdot \Sigma \cdot \epsilon, \qquad (15)$$

So, Eqn. 13 is rewritten as,

$$\underset{\epsilon \in \mathbb{R}^D}{\text{minimize}} \quad ||\epsilon||_2^2, \qquad (16)$$

$$\text{subject to} \quad \Sigma \cdot \epsilon \leq U^\top \cdot \big(\mathbf{b} - A \cdot x^{\text{anchor}}\big),$$

where due to the block diagonal structure of $\Sigma$, we can split $\epsilon = \begin{bmatrix} \epsilon_r \\ \epsilon_o \end{bmatrix}$, $\epsilon_r \in \mathbb{R}^{d-1}$ and $\epsilon_o$ lies in the null space of $A$ and thus can be ignored and set $\epsilon_o = \mathbf{0}$ since we are minimizing $||\epsilon||_2^2$. We further have,

$$\underset{\epsilon_r \in \mathbb{R}^{d-1}}{\text{minimize}} \quad ||\epsilon_r||_2^2, \qquad (17)$$

$$\text{subject to} \quad \Sigma_r \cdot \epsilon_r \leq U^\top \cdot \big(\mathbf{b} - A \cdot x^{\text{anchor}}\big),$$

which is **now a low-dimensional optimization problem with linear constraints,** and can be solved by many QP solvers efficiently, such as the cvxpy package (Diamond & Boyd, 2016; Agrawal et al., 2018). Let $\tilde{\mathbf{b}} = U^\top \cdot \big(\mathbf{b} - A \cdot x^{\text{anchor}}\big)$ which represents the projection of the constraint boundary onto the orthogonal basis formed by the left singular vectors. Then, the constraint $\Sigma_r \cdot \epsilon_r \leq \tilde{\mathbf{b}}$ also shows the feasibility and constraint satisfaction: *the smaller the ratio $\frac{\tilde{\mathbf{b}}_i}{\sigma_i}$ for a singular value $\sigma_i$ ($i = 1, \cdots, d-1$) is, the easier it is to satisfy the corresponding constraint.*

Denote by $\epsilon_r^*$ the optimized solution of Eqn. 17. The optimal solution of $\epsilon$ is $\epsilon^* = \begin{bmatrix} \epsilon_r^* \\ \mathbf{0} \end{bmatrix}$ by definition. Thus, $\delta^* = V \cdot \epsilon^*$. **We can directly recover the optimal solution $x^*$ in the image space** by,

$$x^* = x^{\text{anchor}} + \delta^*,$$

$$= x^{\text{anchor}} + V \cdot \begin{bmatrix} \epsilon_r^* \\ \mathbf{0} \end{bmatrix} = x^{\text{anchor}} + V_r \cdot \epsilon_r^*, \quad (18)$$

which can be understood from the QP perspective in the Appendix C, and is used in updating the perturbation and the perturbed image for the next, $(i+1)$-th iteration of our RisingAttacK,

$$\delta^{(i+1)} = \text{Clamp}(x^*) - x^{\text{benign}}, \quad (19)$$

$$x^{\text{perturb}} = x^{\text{benign}} + \delta^{(i+1)}. \quad (20)$$

### 3.2.3. $\ell_\infty$ PERCENTILE PROJECTION

For the solution $\delta^* = V_r \cdot \epsilon_r^*$ based on Eqn. 17, we observe that it often exhibits disproportionately high $\ell_\infty$ norms. We observe this large $\ell_\infty$ is driven by very few components (pixel values) of our solution and the overall quality of our solution is not contaminated by these extreme values (or outliers). We hypothesize that the outliers might be caused by the first-order Taylor linearization that is not sufficiently accurate at those pixels. To alleviate this issue, we resort to a $\ell_\infty$ Percentile Projection as the post-processing step. Specifically, we compute

$$\tau = \text{Percentile}(|\delta^*|, 0.995) \quad (21)$$

where $\tau$ indicates 99.5th percentile of magnitudes in our solution. We then element-wisely project $\delta^*$ to this percentile,

$$\delta_i^* \leftarrow \text{Sign}(\delta_i^*) \times \min(|\delta_i^*|, \tau), \quad (22)$$

where $i$ is the entry index.

### 3.2.4. ANCHOR POINT SELECTION

When the number of iterations is infinite (or very high), choosing $x^{\text{anchor}} = x^{\text{benign}}$ in Eqn. 11 yields the lowest energy solution upon convergence. This is because each step of the optimization directly minimizes the distance from $x$ to $x^{\text{benign}}$, aligning the solution trajectory with the global objective. However, in practice, the number of iterations is limited, and $x^{\text{benign}}$ does not lie within the adversarial region during intermediate iterations. As a result, only using $x^{\text{benign}}$ as the anchor point can significantly delay reaching the adversarial region, especially when the constraint set is complex (when $K$ is large).

On the other hand, choosing $x^{\text{perturb}}$ (the current perturbed image), as the anchor point ensures rapid progress toward the adversarial region. Since the optimization minimizes the distance from $x$ to $x^{\text{perturb}}$ at each step, the solution quickly adjusts to satisfy the constraints. However, this may lead to suboptimal solutions in terms of perturbation energy,

as the optimization prioritizes feasibility over minimizing perturbation energy.

**Alternating Anchor Point Strategy.** To balance the trade-offs between rapid feasibility and minimal energy, we implement an alternating anchor point strategy. This approach dynamically switches between $x^{\text{benign}}$ and $x^{\text{perturb}}$ as the anchor point based on the current optimization state.

- If the number of iterations since the last feasible solution exceeds $\beta$ (a predefined threshold), we set $x^{\text{anchor}} = x^{\text{perturb}}$ to prioritize reaching the adversarial region.
- Otherwise, we set $x^{\text{anchor}} = x^{\text{benign}}$ to continue minimizing the perturbation energy while staying within the adversarial region.

### 3.2.5. INTERPRETATION OF RISINGATTACK

Eqn. 18 provides an intuitive interpretation for the optimized perturbation $\delta^* = V_r \cdot \epsilon_r^*$ at each iteration of the optimization. The perturbation is the learned linear combination with coefficients in $\epsilon_r^* \in \mathbb{R}^{d-1}$ of $d-1$ image bases, i.e., columns in $V_r \in \mathbb{R}^{D \times (d-1)}$. Recall that each column in $V_r$ represents a principle direction in the image space that can affect logit ranking the most subject to how large the corresponding singular value is. The learned weighted sum of the columns of $V_r$ can provide most efficient perturbation, as shown by the consistently smaller perturbation energy obtained in our experiments.

**Potential Defensive Insights.** By analyzing which singular vectors in $V_r$ correspond to large singular values, defensive strategies can be developed by reinforcing robustness in those vulnerable directions against ordered top-$K$ attacks. Meanwhile, adversarial training can be guided to target these critical subspaces. We leave those for future work.

## 4. Experiments

In this section, we evaluate our RisingAttacK in the ImageNet-1k benchmark (Russakovsky et al., 2015), and compare with QuadAttacK (Paniagua et al., 2023).

*Models Under AttacK.* Following QuadAttacK, we use two representative ConvNets (ResNet-50 (He et al., 2016) and DenseNet-121 (Huang et al., 2017)) and two Vision Transformers (ViT-B (Dosovitskiy et al., 2020) and DEiT-B (Touvron et al., 2021)). Their ImageNet-1k pretrained checkpoints are from the *timm* package (Wightman, 2019).

*Data and Attack Targets.* We use ImageNet-1k `val` images from which we select and sample a subset consisting of class-balanced 1000 images (i.e., one image per class). The 1000 benign images can be correctly classified by all the four models. For each image, five ordered target sets are randomly sampled for each value of $K$ ($K = 1, 5, 10, 15, 20, 25, 30$), see Appendix D for details.

*Table 1.* **Ordered top-$K$ attack results averaged across 5 different seeds**. Overall, our RisingAttacK shows a big leap forward in advancing ordered top-$K$ attacks, outperforming the prior state-of-the-art method, QuadAttacK (Paniagua et al., 2023) by a large margin in most cases (higher ASRs with lower $\ell_p$ norms). $\ell_\infty$-norms in red is to show they are treated as being "visually imperceptible" based on the commonly used threshold $8/255 = 0.0314$. The subscripts of methods (30 and 60) represent the computing budgets.

(a) ResNet-50 (He et al., 2016)

| Top-$K$ | Method | ASR ↑ | $\ell_1$ ↓ | $\ell_2$ ↓ | $\ell_\infty$ ↓ | Time (s/img) ↓ | FoM ↑ |
|---|---|---|---|---|---|---|---|
| Top-30 | QuadAttacK₆₀ | 0.2076 | 11.8070 | 3654.9139 | 0.1349 | **3.3947** | 6.4793 |
| | RisingAttacK₆₀ | **0.6642** | **7.0271** | **2081.8960** | **0.0511** | 17.0013 | |
| | QuadAttacK₃₀ | Failed | | | | **1.6539** | inf |
| | RisingAttacK₃₀ | 0.0022 | 6.2378 | 1844.3013 | 0.0470 | 8.5619 | |
| Top-25 | QuadAttacK₆₀ | 0.6018 | 11.6214 | 3599.8101 | 0.1301 | **3.4167** | 3.6439 |
| | RisingAttacK₆₀ | **0.8420** | **5.2960** | **1561.6462** | **0.0393** | 14.0839 | |
| | QuadAttacK₃₀ | 0.0018 | 10.4263 | 3259.2773 | 0.0991 | **1.7058** | 48.9628 |
| | RisingAttacK₃₀ | **0.0392** | **5.1218** | **1511.3347** | **0.0388** | 7.0999 | |
| Top-20 | QuadAttacK₆₀ | 0.8344 | 10.0891 | 3133.6199 | 0.1079 | **3.4039** | 3.1100 |
| | RisingAttacK₆₀ | 0.8306 | **3.7474** | **1101.1521** | **0.0281** | 6.7267 | |
| | QuadAttacK₃₀ | 0.0978 | 9.0948 | 2850.0433 | 0.0858 | **1.7264** | 1.9481 |
| | RisingAttacK₃₀ | **0.0666** | **3.4854** | **1022.5585** | **0.0269** | 3.7216 | |
| Top-15 | QuadAttacK₆₀ | 0.9440 | 8.3368 | 2600.7510 | 0.0822 | **3.4839** | 3.2229 |
| | RisingAttacK₆₀ | **0.9868** | **3.0150** | **878.9222** | **0.0233** | 5.1634 | |
| | QuadAttacK₃₀ | 0.4922 | 7.8296 | 2451.8036 | 0.0717 | **1.7382** | 3.3674 |
| | RisingAttacK₃₀ | **0.5856** | **2.9944** | **873.3877** | **0.0234** | 2.8794 | |
| Top-10 | QuadAttacK₆₀ | 0.9866 | 6.5228 | 2044.5753 | 0.0576 | 3.7396 | 3.3482 |
| | RisingAttacK₆₀ | **0.9936** | **2.0825** | **602.1784** | **0.0167** | **3.3991** | |
| | QuadAttacK₃₀ | 0.8460 | 6.3547 | 1994.8023 | 0.0544 | **1.7593** | 2.9244 |
| | RisingAttacK₃₀ | 0.8064 | **2.1748** | **630.0922** | **0.0175** | 1.7965 | |
| Top-5 | QuadAttacK₆₀ | 0.9968 | 4.0029 | 1261.2314 | 0.0309 | 4.5257 | 3.3373 |
| | RisingAttacK₆₀ | 0.9558 | **1.1534** | **330.1495** | **0.0098** | **1.8225** | |
| | QuadAttacK₃₀ | 0.9590 | 3.9539 | 1246.4929 | 0.0300 | 2.1458 | 2.6681 |
| | RisingAttacK₃₀ | 0.9504 | **1.4693** | **420.0254** | 0.0124 | **0.9517** | |
| Top-1 | QuadAttacK₆₀ | 0.9996 | 1.4443 | 467.1178 | 0.0083 | 5.3373 | 2.1564 |
| | RisingAttacK₆₀ | 0.9992 | **0.6144** | **165.8517** | **0.0064** | **0.6114** | |
| | QuadAttacK₃₀ | 0.9772 | 1.4244 | 461.1199 | 0.0080 | 2.6411 | 1.4638 |
| | RisingAttacK₃₀ | **0.9986** | **0.9155** | **251.6174** | 0.0088 | **0.3201** | |

(b) DenseNet-121 (Huang et al., 2017)

| Top-$K$ | Method | ASR ↑ | $\ell_1$ ↓ | $\ell_2$ ↓ | $\ell_\infty$ ↓ | Time (s/img) ↓ | FoM ↑ |
|---|---|---|---|---|---|---|---|
| Top-30 | QuadAttacK₆₀ | Failed | | | | **4.5409** | inf |
| | RisingAttacK₆₀ | 0.4074 | 14.7263 | 4393.8482 | 0.1051 | 20.3156 | |
| | QuadAttacK₃₀ | Failed | | | | **2.3266** | 0 |
| | RisingAttacK₃₀ | Failed | | | | 10.2335 | |
| Top-25 | QuadAttacK₆₀ | 0.1734 | 13.1825 | 4053.5759 | 0.1531 | **4.1657** | 8.5496 |
| | RisingAttacK₆₀ | **0.9370** | **9.9898** | **2945.3574** | **0.0747** | 16.8643 | |
| | QuadAttacK₃₀ | Failed | | | | **2.2016** | inf |
| | RisingAttacK₃₀ | 0.1094 | 9.9203 | 2921.6770 | 0.0756 | 8.5279 | |
| Top-20 | QuadAttacK₆₀ | 0.8340 | 11.6266 | 3583.3589 | 0.1268 | **4.0066** | 2.6290 |
| | RisingAttacK₆₀ | **0.9812** | **5.9921** | **1744.8239** | **0.0468** | 8.2901 | |
| | QuadAttacK₃₀ | 0.0330 | 9.8564 | 3072.6790 | 0.0923 | **2.0206** | 23.7723 |
| | RisingAttacK₃₀ | **0.4500** | **6.1377** | **1786.5613** | **0.0485** | 4.6070 | |
| Top-15 | QuadAttacK₆₀ | 0.9866 | 9.2713 | 2884.2755 | 0.0887 | **3.8963** | 2.3310 |
| | RisingAttacK₆₀ | **1.0000** | **4.3657** | **1252.9889** | **0.0359** | 6.2878 | |
| | QuadAttacK₃₀ | 0.5088 | 8.6281 | 2697.9823 | 0.0771 | **1.8919** | 3.5524 |
| | RisingAttacK₃₀ | **0.9362** | **4.7380** | **1362.7350** | **0.0387** | 3.5501 | |
| Top-10 | QuadAttacK₆₀ | 0.9986 | 6.7558 | 2123.4894 | 0.0545 | **3.8256** | 2.5458 |
| | RisingAttacK₆₀ | **1.0000** | **2.6903** | **759.1986** | **0.0235** | 4.2223 | |
| | QuadAttacK₃₀ | 0.9392 | 6.6701 | 2098.0095 | 0.0531 | **1.8918** | 2.4272 |
| | RisingAttacK₃₀ | **0.9880** | **2.9210** | **827.1606** | **0.0253** | 2.2937 | |
| Top-5 | QuadAttacK₆₀ | 0.9998 | 3.9671 | 1258.1706 | 0.0264 | 3.8644 | 3.0870 |
| | RisingAttacK₆₀ | 0.9994 | **1.2169** | **331.6714** | **0.0119** | **2.2643** | |
| | QuadAttacK₃₀ | 0.9924 | 3.9526 | 1253.5745 | 0.0262 | 1.8502 | 2.2794 |
| | RisingAttacK₃₀ | **0.9982** | **1.6603** | **457.9204** | **0.0156** | **1.2082** | |
| Top-1 | QuadAttacK₆₀ | 1.0000 | 1.5191 | 503.0047 | 0.0070 | 3.0413 | 1.9466 |
| | RisingAttacK₆₀ | 1.0000 | **0.7001** | **177.1356** | **0.0085** | **0.8046** | |
| | QuadAttacK₃₀ | 0.9960 | 1.5144 | 501.4779 | 0.0070 | 1.5519 | 1.2739 |
| | RisingAttacK₃₀ | **1.0000** | **1.0708** | **280.0300** | 0.0116 | **0.4255** | |

(c) ViT-B (Dosovitskiy et al., 2020)

| Top-$K$ | Method | ASR ↑ | $\ell_1$ ↓ | $\ell_2$ ↓ | $\ell_\infty$ ↓ | Time (s/img) ↓ | FoM ↑ |
|---|---|---|---|---|---|---|---|
| Top-30 | QuadAttacK₆₀ | 0.3272 | 9.6708 | 2938.2587 | 0.1032 | **5.2135** | 3.1589 |
| | RisingAttacK₆₀ | **0.9534** | 9.7262 | **2721.2876** | **0.0876** | 43.3954 | |
| | QuadAttacK₃₀ | Failed | | | | **2.7870** | inf |
| | RisingAttacK₃₀ | **0.5568** | 11.4132 | 3206.4565 | 0.1029 | 21.7179 | |
| Top-25 | QuadAttacK₆₀ | 0.6872 | 9.4331 | 2860.6667 | 0.1002 | **5.2723** | 2.6425 |
| | RisingAttacK₆₀ | **0.9944** | **5.5706** | **1520.5703** | **0.0526** | 36.0486 | |
| | QuadAttacK₃₀ | Failed | | | | **2.7354** | inf |
| | RisingAttacK₃₀ | **0.7536** | 7.7050 | 2126.3211 | 0.0721 | 18.0775 | |
| Top-20 | QuadAttacK₆₀ | 0.7828 | 7.9108 | 2393.0875 | 0.0815 | **5.0069** | 2.8308 |
| | RisingAttacK₆₀ | **0.9864** | **3.7609** | **1007.6887** | **0.0360** | 15.8230 | |
| | QuadAttacK₃₀ | 0.0004 | 6.3770 | 1992.7502 | 0.0533 | **2.6210** | 1610.8632 |
| | RisingAttacK₃₀ | **0.4956** | **4.9482** | **1343.2135** | **0.0473** | 7.9615 | |
| Top-15 | QuadAttacK₆₀ | 0.8404 | 6.2661 | 1893.5173 | 0.0620 | **4.7622** | 2.7231 |
| | RisingAttacK₆₀ | **0.9988** | **2.8751** | **753.1852** | **0.0284** | 11.9841 | |
| | QuadAttacK₃₀ | 0.0056 | 4.7982 | 1495.8188 | 0.0385 | **2.4245** | 164.8583 |
| | RisingAttacK₃₀ | **0.7510** | **3.8944** | **1038.7394** | **0.0379** | 6.0305 | |
| Top-10 | QuadAttacK₆₀ | 0.9130 | 4.5246 | 1374.2282 | 0.0410 | **4.6368** | 2.5247 |
| | RisingAttacK₆₀ | **0.9936** | **1.9915** | **508.8791** | **0.0288** | 8.2583 | |
| | QuadAttacK₃₀ | 0.0252 | 3.4999 | 1094.6987 | 0.0261 | **2.3034** | 36.7947 |
| | RisingAttacK₃₀ | **0.7112** | **2.6247** | **684.1576** | 0.0267 | 4.1602 | |
| Top-5 | QuadAttacK₆₀ | **0.9980** | 6.6439 | 1128.3054 | 0.0288 | **4.3981** | 1.7630 |
| | RisingAttacK₆₀ | 0.5712 | **1.1650** | **292.6494** | **0.0128** | 4.4038 | |
| | QuadAttacK₃₀ | 0.5024 | 3.2930 | 1029.8490 | 0.0242 | **2.1108** | 2.3688 |
| | RisingAttacK₃₀ | **0.5980** | **1.6101** | **406.4644** | **0.0174** | 2.2197 | |
| Top-1 | QuadAttacK₆₀ | 0.9998 | 1.5736 | 509.7575 | 0.0081 | 2.6007 | 3.2121 |
| | RisingAttacK₆₀ | 0.9388 | **0.4365** | **96.0745** | **0.0060** | **1.2715** | |
| | QuadAttacK₃₀ | 0.9958 | 1.5681 | 508.0591 | 0.0081 | 1.3040 | 2.1102 |
| | RisingAttacK₃₀ | 0.9362 | **0.6578** | **149.1661** | 0.0086 | **0.6417** | |

(d) DEiT-B (Touvron et al., 2021)

| Top-$K$ | Method | ASR ↑ | $\ell_1$ ↓ | $\ell_2$ ↓ | $\ell_\infty$ ↓ | Time (s/img) ↓ | FoM ↑ |
|---|---|---|---|---|---|---|---|
| Top-30 | QuadAttacK₆₀ | 0.0640 | 9.3734 | 2860.9240 | 0.0997 | **4.1792** | 8.8333 |
| | RisingAttacK₆₀ | **0.5150** | 9.4432 | **2697.9176** | **0.0804** | 43.3521 | |
| | QuadAttacK₃₀ | Failed | | | | **2.3032** | inf |
| | RisingAttacK₃₀ | **0.0600** | 11.0771 | 3165.6910 | 0.0957 | 21.6930 | |
| Top-25 | QuadAttacK₆₀ | 0.8644 | 9.3780 | 2849.8222 | 0.0960 | **4.0966** | 2.1975 |
| | RisingAttacK₆₀ | **0.9854** | **5.1921** | **1434.6160** | **0.0482** | 36.1084 | |
| | QuadAttacK₃₀ | Failed | | | | **2.2173** | inf |
| | RisingAttacK₃₀ | **0.6748** | 6.3220 | 1763.4334 | 0.0581 | 18.1108 | |
| Top-20 | QuadAttacK₆₀ | 0.9612 | 7.6974 | 2343.5441 | 0.0735 | **4.1868** | 2.8466 |
| | RisingAttacK₆₀ | **0.9956** | **2.9174** | **781.2607** | **0.0282** | 15.8331 | |
| | QuadAttacK₃₀ | 0.0032 | 6.2491 | 1950.0525 | 0.0524 | **2.1503** | 325.7059 |
| | RisingAttacK₃₀ | **0.6348** | **3.8373** | **1045.3953** | **0.0366** | 7.9624 | |
| Top-15 | QuadAttacK₆₀ | 0.9750 | 6.0671 | 1852.4958 | 0.0544 | **3.9525** | 2.8819 |
| | RisingAttacK₆₀ | **1.0000** | **2.2015** | **573.3811** | **0.0223** | 11.9810 | |
| | QuadAttacK₃₀ | 0.0338 | 4.9874 | 1558.1460 | 0.0386 | **2.0234** | 42.8983 |
| | RisingAttacK₃₀ | **0.9278** | **3.1490** | **838.2295** | **0.0310** | 6.0263 | |
| Top-10 | QuadAttacK₆₀ | 0.9762 | 4.3693 | 1346.6326 | 0.0353 | **3.8755** | 2.9455 |
| | RisingAttacK₆₀ | **0.9996** | **1.5076** | **379.6582** | **0.0162** | 8.2610 | |
| | QuadAttacK₃₀ | 0.1298 | 3.5782 | 1123.3760 | 0.0256 | **1.9552** | 11.4300 |
| | RisingAttacK₃₀ | **0.9200** | **2.1465** | **556.2741** | **0.0222** | 4.1613 | |
| Top-5 | QuadAttacK₆₀ | 0.9984 | 3.3975 | 1064.7252 | 0.0243 | **3.4381** | 3.1378 |
| | RisingAttacK₆₀ | **0.9992** | **1.0575** | **254.5953** | **0.0121** | 4.4027 | |
| | QuadAttacK₃₀ | 0.7794 | 3.2526 | 1024.0607 | 0.0225 | **1.7718** | 2.6286 |
| | RisingAttacK₃₀ | **0.8800** | **1.3450** | **334.9398** | **0.0149** | 2.2165 | |
| Top-1 | QuadAttacK₆₀ | 1.0000 | 1.3910 | 459.6084 | 0.0063 | 2.9955 | 3.9437 |
| | RisingAttacK₆₀ | 0.9794 | **0.3340** | **68.5738** | **0.0052** | **1.2708** | |
| | QuadAttacK₃₀ | **0.9994** | 1.3899 | 459.3060 | 0.0063 | 1.4404 | 2.4502 |
| | RisingAttacK₃₀ | 0.9772 | **0.5249** | **114.2980** | 0.0073 | **0.6426** | |

*Metrics.* The metrics used to evaluate the attack methods include the Attack Success Rate (ASR), as well as the $\ell_1$, $\ell_2$, and $\ell_\infty$ norms of the perturbations. ASR quantifies the fraction of adversarial examples satisfying the ordered top-$K$ constraints (larger is better). $\ell_p$ norms are computed based on successful adversarial examples (lower is better, indicating less visually-perceptible). We note that $\ell_p$ norms are compatible between different methods only when their ASRs are similar. For example, a method may show very low $\ell_p$ norms when the ASR is also very low (i.e., it can only

attack a few images). To compare the relative improvement of one method (with $\mathrm{ASR}^1$ and $\ell_p^1$ norms) against another one (with $\mathrm{ASR}^2$ and $\ell_p^2$ norms), we propose to use a holistic figure of merits (FoM),

$$\mathrm{FoM} = \frac{\mathrm{ASR}^1}{\mathrm{ASR}^2} \cdot \frac{1}{3} \cdot \sum_{p \in \{1, 2, \infty\}} \frac{\ell_p^2}{\ell_p^1}, \qquad (23)$$

where when the opponent method fails, i.e., $\mathrm{ASR}^2 = 0$, we set FoM$= +\infty$. Similarly, we set FoM$= -\infty$ if the primary method fails while the opponent method succeeds,

and FoM= 0 if both methods fail. When the FoM> 1, we say the primary method is holistically better than the opponent method. We report the `Mean` metrics across the five sampled targets for each $K$. We also adopt the commonly used $\ell_\infty = 8/255$ as the threshold to characterize the "visual imperceptibility" of learned adversarial perturbations (Croce et al., 2020). See Appendix E for details of metrics including `Best,` `Mean` and `Worst` comparisons.

*Baselines.* We mainly compare with QuadAttacK (Paniagua et al., 2023) since it is the prior state-of-the-art method, significantly outperforming the $CW^K$ and AD (Adversarial Distillation) (Zhang & Wu, 2020). For a fair comparison, both methods are tested under identical experimental conditions. For all experiments, each attack is evaluated at 30 and 60 optimization iterations to analyze its performance under varying computational budgets. The initial perturbation for all attacks is set to zero, ensuring consistent starting conditions across methods.

*Results and Analyses.* **Our proposed RisingAttacK shows a big leap forward in advancing ordered top-$K$ attacks, which in turn verifies the significant advantages of learning attacks directly in the image space by our proposed SQP formulation.** Results of ordered top-$K$ attacks for the four models are shown in Table 1(a), 1(b), 1(c) and 1(d).

- Based on the FoM evaluation (Eqn. 23), our RisingAttacK consistently outperforms the previous state-of-the-art method, QuadAttacK across all $K$ (=1,5,10,20, 25,30) and all four models. It achieves FoMs greater than 2 in most cases (i.e., holistically 2x better than QuadAttacK).
- Our RisingAttacK facilitates learning visually-imperceptible perturbations up to $K = 20$ for ResNet50 and DEiT-B, $K = 15$ for ViT-B, and $K = 10$ for DenseNet121, based on the $\ell_\infty$ threshold, significantly outperforming QuadAttacK.

Fig. 2 show examples of learned adversarial examples and perturbations using RisingAttacK$_{60}$. More examples are provided in the Appendix F.

*More Results.* We also show results of using the lowest-K predictions of each benign image by each model as the ordered top-K attack targets (Table 6 in the Appendix E). Ordered top-K targets by this image- and model-specific selection method are intuitively deemed as more difficult to attack, as empirically shown in (Zhang & Wu, 2020). Counterintuitively, our results show they are not more difficult than randomly sampled targets using both QuadAttacK and our RisingAttacK.

*The Average Speed (second/image).* We note that for $K = 1$ our RisingAttacK is consistently faster than QuadAttacK. For $K > 1$, QuadAttacK is mostly faster than our RisingAttacK. The main reason is due to the current implementation of computing the logit-to-image Jacobian matrix in PyTorch,

for which we used PyTorch 2.6 and the `jacrev` and `vmap` (with chunk size 100) functions in the `torch.func` library. When $K$ is larger than 1, based on Eqn. 9, we maintain $K + M + 1$ logits with $M = 5 \cdot K$. We did not test other factors for $M$ (e.g., $2 \cdot K$ or a predefined constant such as 5). We will address this speed limitation in future work.

## 5. Conclusion

This paper presents RisingAttack, a novel method for learning ordered top-$K$ targeted white-box adversarial attacks by directly solving the non-linearly constrained optimization problem in image space under the sequential quadratic programming framework. Our RisingAttacK provides a simple yet elegant solution: ordered top-$K$ adversarial perturbations can be learned via iteratively optimizing linear combinations of the right singular vectors (corresponding to non-zero singular values) of the attack-targets-ranking constrained logit-to-image Jacobian matrix. Through experiments on four ImageNet-1k trained DNNs, our RisingAttacK shows a big leap forward in advancing ordered top-$K$ attacks in terms of a proposed figure-of-merits metric, significantly outperforming the previous state-of-the-art method, QuadAttacK.

## Impact Statement

This work advances the field of adversarial machine learning by introducing RisingAttacK. By improving the efficiency and scalability of ordered top-$K$ adversarial attacks, particularly for large $K$ values, this research highlights critical vulnerabilities in modern DNNs. However, adversarial attack methods also pose risks, as they may be misused to compromise real-world systems. For example, attacks on ranking-based systems could be exploited to manipulate search engine results or recommendation algorithms. To mitigate these risks, this work should be viewed as a tool for potentially strengthening defenses (e.g., as critics for them) rather than enabling malicious use. In addition, this work contributes to the broader exploration of optimization in machine learning by integrating techniques from traditional nonlinear programming into neural network-based problems. This direction holds promise for both adversarial research and other optimization tasks in machine learning, offering a foundation for solving increasingly complex challenges.

## Acknowledgments

This research is partly supported by NSF IIS-1909644, ARO Grant W911NF1810295, ARO Grant W911NF2210010, NSF CMMI-2024688 and NSF IUSE-2013451. The views and conclusions contained herein are those of the authors and should not be interpreted as necessarily representing the official policies or endorsements, either expressed or implied, of ARO, NSF or the U.S. Government.

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

## A. Background on QuadAttacK

QuadAttacK (Paniagua et al., 2023) addresses the challenge of optimizing Eqn. 4 by first "lifting" it into the feature space, i.e., the output space of $f(\cdot)$, see the left-bottom of Fig 1. At a given iteration $i$, let $\delta^{(i)}$ be the current perturbation, and $x^{\text{perturb}} = x^{\text{benign}} + \delta^{(i)}$ the current perturbed image with $z^{\text{perturb}} = f(x^{\text{perturb}})$ its DNN features. QuadAttacK aims to iteratively find the optimal perturbed features $z$ around $z^{\text{perturb}}$ to satisfy the constraints by,

$$\underset{z}{\text{minimize}} \quad ||z - z^{\text{perturb}}||_2^2, \tag{24}$$

$$\text{subject to} \quad \mathbb{K} \cdot (W \cdot z + b) > 0,$$

where the nonlinear backbone $f(\cdot)$ is eliminated from the constraints. Eqn. 24 can be solved by a QP package (Amos & Kolter, 2017). With the optimized $z^*$, the adversarial perturbation is updated by back-propagating the feature distance to the image space through the highly non-linear DNN backbone $f(\cdot)$,

$$\delta^{(i+1)} = \delta^{(i)} - \gamma \cdot \frac{\partial}{\partial \delta} \big( \lambda \cdot ||z^* - z^{\text{perturb}}||_2^2 + ||\delta||_p \big)|_{\delta = \delta^{(i)}}, \tag{25}$$

where $\gamma$ is the learning rate, and $\lambda$ the trade-off parameter between feature distance and image perturbation. The perturbed image is updated by,

$$x^{\text{perturb}} = \text{Clamp}(x^{\text{benign}} + \delta^{(i+1)}). \tag{26}$$

QuadAttacK is executed iteratively with respect to a predefined computing budget (e.g., 30 or 60 iterations). As aforementioned, there is a gap between the optimized $z^*$ (Eqn. 24) in the feature space and the computed $\delta^{(i+1)}$ (Eqn. 25) in the image space in terms of satisfying the ordered top-K constraints, which leads to suboptimal adversarial examples (Eqn. 26).

## B. Details on Compact Ordered Top-K Constraints

In Sec. 3.2.1, we introduce the mapping $G$ (Eqn. 9) that reorders the logits and augments them with a differentiable nonlinear term, reproduced here,

$$G : \ell(\cdot) \in \mathbb{R}^C \to \mathbf{l}(\cdot) \in \mathbb{R}^{d=K+M+1},$$

where $K = |\mathcal{T}|$ is the number of attack targets, $M$ is the number of highest non-target logits to include explicitly (e.g., $M = 5 \cdot K$), and the final term is the soft-maximum of the remaining logits. We have,

- *The ordered top-$K$ targets:* $\mathbf{l}(x)_i = \ell(x)_{t_i}$, for $i \in \{1, \dots, K\}$, where $t_i \in \mathcal{T}$ is the $i$-th target class. These targets remain the same during the optimization.
- *The ordered top-$M$ non-targets:* $\mathbf{l}(x)_{K+j} = \ell(x)_{m_j}$, where $j \in [1, \cdots, M]$, and $m_j = \arg\text{sort}_j\{\ell(x)_i; i \in \mathcal{Y} \setminus \mathcal{T}\}$, i.e., $\ell(x)_{m_j}$ is the $j$-th largest non-target logit. For example, $M = 5 \times K$. Denote by $\mathcal{M}$ the ordered top-$M$ non-target classes, which are dynamic during the optimization.
- *The Soft-Maximum of logits of the Remaining Classes:* $\mathbf{l}(x)_d = \text{SmoothMax}\Big( \{\ell(x)_j; j \in \mathcal{Y} \setminus (\mathcal{T} \cup \mathcal{M})\} \Big)$, where $d = K + M + 1$, and $\text{SmoothMax}(\cdot)$ is differentiable and enables gradient distribution (rather than switching) in learning, which is defined by,

$$\text{SmoothMax}(v) = \text{Sum}(\text{Softmax}(v) \odot v), \tag{27}$$

  where $\odot$ represents element-wise (Hadamard) product. It is straightforward to show that $\text{mean}(v) \le \text{SmoothMax}(v) \le \text{max}(v)$ for any real vectors $v$.

**We note that the inclusion of the top-$M$ non-target logits** is to ensure that the compact constraints remain robust, even in cases where the $\text{SmoothMax}(\cdot)$ function introduces significant nonlinearity.

## C. QP for Recovering Perturbation in the Image Space

We show the solution (Eqn. 18) can be understood from the QP perspective. Based on $\delta = V \cdot \epsilon$ and $\delta = x - x^{\text{anchor}}$, we have,

$$\epsilon = V^\top \cdot (x - x^{\text{anchor}}), \tag{28}$$

$$\epsilon_r = V_r^\top \cdot x - V_r^\top \cdot x^{\text{anchor}} \triangleq x_r - x_r^{\text{anchor}}, \tag{29}$$

where $x_r$ is the projection of $x$, and $x_r^{\text{anchor}}$ the projection of the anchor image.

Minimizing $||\epsilon_r||_2^2$ is to find the optimal $x_r^*$ that is closest to $x_r^{\text{anchor}}$. We have, $x_r^* = x_r^{\text{anchor}} + \epsilon_r^*$, with which the QP for

recovering the optimal perturbation in the original image space is to,

$$\underset{x \in [0,1]^D}{\text{minimize}} \quad ||x - x^{\text{anchor}}||_2^2, \tag{30}$$

$$\text{subject to} \quad V_r^\top \cdot x = x_r^*,$$

which only involves equality constraints, making it computationally efficient to solve, even though $x$ is in the high-dimensional image space. We show that Eqn. 30 has a closed-form solution, reproducing the result in Eqn. 18.

Recall $\epsilon_r^*$ is the solution from solving Eqn. 17, $x_r^{\text{anchor}} = V_r^\top \cdot x^{\text{anchor}}$ and $x_r^* = x_r^{\text{anchor}} + \epsilon_r^*$. Eqn. 30 is reproduced here,

$$\underset{x \in [0,1]^D}{\text{minimize}} \quad ||x - x^{\text{anchor}}||_2^2, $$

$$\text{subject to} \quad V_r^\top \cdot x = x_r^*,$$

which can be re-expressed by expanding the objective function and removing the constant term as,

$$\underset{x \in [0,1]^D}{\text{minimize}} \quad x^\top \cdot x - 2 \cdot x^{\text{anchor}^\top} \cdot x, \tag{31}$$

$$\text{subject to} \quad V_r^\top \cdot x = x_r^*,$$

And the Lagrangian is,

$$\mathcal{L}(x, \lambda) = x^\top \cdot x - 2 \cdot x^{\text{anchor}^\top} \cdot x + \lambda^\top \cdot (V_r^\top \cdot x - x_r^*). \tag{32}$$

We obtain estimating equations from the derivatives as follows,

$$\frac{\partial}{\partial x} \mathcal{L}(x, \lambda) = 2 \cdot x - 2 \cdot x^{\text{anchor}} + V_r \cdot \lambda = 0, \tag{33}$$

$$\Rightarrow \quad x = x^{\text{anchor}} - \frac{1}{2} V_r \cdot \lambda, \tag{34}$$

$$\frac{\partial}{\partial \lambda} \mathcal{L}(x, \lambda) = V_r^\top \cdot x - x_r^* = 0, \tag{35}$$

$$\Rightarrow \quad V_r^\top \cdot x = x_r^*, \tag{36}$$

We have,

$$V_r^\top \cdot (x^{\text{anchor}} - \frac{1}{2} V_r \cdot \lambda) = x_r^* \tag{37}$$

$$\Rightarrow \quad \lambda = 2 \cdot (V_r^\top \cdot x^{\text{anchor}} - x_r^*) = 2 \cdot \left( x_r^{\text{anchor}} - (x_r^{\text{anchor}} + \epsilon_r^*) \right) = -2 \cdot \epsilon_r^*, \tag{38}$$

So, we have,

$$x = x^{\text{anchor}} - \frac{1}{2} V_r \cdot (-2 \cdot \epsilon_r^*) = x^{\text{anchor}} + V_r \cdot \epsilon_r^*, \tag{39}$$

which reproduces Eqn. 18.

## D. Details of Attack Targets

We use 5 random seeds (42, 52, 62, 72 and 82) and sample 5 lists of ordered top-30 targets as follows:

- *seed=42*: (643): mask, (409): analog-clock, (798): slide-rule, (250): Siberian-husky, (593): harmonica, (47): African-chameleon, (142): dowitcher, (276): hyena, (908): wing, (721): pillow, (57): garter-snake, (257): Great-Pyrenees, (397): puffer, (954): banana, (203): West-Highland-white-terrier, (172): whippet, (294): brown-bear, (803): snowplow, (76): tarantula, (811): space-heater, (817): sports-car, (608): jean, (977): sandbar, (711): perfume, (157): papillon, (51): triceratops, (428): barrow, (84): peacock, (531): digital-watch, (478): carton

- *seed=52*: (523): crutch, (330): wood-rabbit, (743): prison, (611): jigsaw-puzzle, (613): joystick, (810): space-bar, (634): lumbermill, (203): West-Highland-white-terrier, (217): English-springer, (816): spindle, (926): hot-pot, (275): African-hunting-dog, (337): beaver, (33): loggerhead, (264): Cardigan, (862): torch, (755): radio-telescope, (949): strawberry, (162): beagle, (488): chain, (251): dalmatian, (292): tiger, (440): beer-bottle, (638): maillot, (722): ping-pong-ball, (349): bighorn, (592): hard-disc, (409): analog-clock, (584): hair-slide, (701): parachute

- *seed=62*: (45): Gila-monster, (224): groenendael, (274): dhole, (54): hognose-snake, (759): reflex-camera, (931):

bagel, (1): goldfish, (478): carton, (51): triceratops, (649): megalith, (117): chambered-nautilus, (652): military-uniform, (601): hoopskirt, (571): gas-pump, (520): crib, (221): Irish-water-spaniel, (869): trench-coat, (102): echidna, (14): indigo-bunting, (670): motor-scooter, (975): lakeside, (511): convertible, (8): hen, (840): swab, (156): Blenheim-spaniel, (928): ice-cream, (24): great-grey-owl, (567): frying-pan, (668): mosque, (866): tractor

- *seed=72*: (678): neck-brace, (329): sea-cucumber, (731): plunger, (829): streetcar, (565): freight-car, (628): liner, (331): hare, (376): proboscis-monkey, (787): shield, (622): lens-cap, (402): acoustic-guitar, (225): malinois, (487): cellular-telephone, (858): tile-roof, (94): hummingbird, (991): coral-fungus, (808): sombrero, (95): jacamar, (649): megalith, (35): mud-turtle, (215): Brittany-spaniel, (246): Great-Dane, (222): kuvasz, (88): macaw, (586): half-track, (424): barbershop, (553): file, (302): ground-beetle, (363): armadillo, (793): shower-cap

- *seed=82*: (280): grey-fox, (942): butternut-squash, (457): bow-tie, (810): space-bar, (811): space-heater, (388): giant-panda, (121): king-crab, (974): geyser, (432): bassoon, (969): eggnog, (633): loupe, (399): abaya, (438): beaker, (329): sea-cucumber, (563): fountain-pen, (661): Model-T, (552): feather-boa, (256): Newfoundland, (859): toaster, (539): doormat, (949): strawberry, (157): papillon, (410): apiary, (569): garbage-truck, (496): Christmas-stocking, (207): golden-retriever, (591): handkerchief, (806): sock, (372): baboon, (219): cocker-spaniel

We use those targets sequentially for $K = 1, 5, 10, 15, 20, 25, 30$ for the four models. The targets are shared by the 1000 testing images. For each testing image, if its ground-truth label is in any ordered top-K targets, we replace it with a different randomly sampled targets.

In addition to the randomly sampled targets, we also test a special case in which the lowest-K predictions by a model for a benign image are used as the ordered top-K attack targets (i.e., the first target is the class of the lowest logit for the benign image, and so far so on). The results are shown in Table 6 in the Appendix E.

# E. Details of Metrics and Full Results

We report the `Mean` metrics (ASRs and $\ell_p$ norms) in the paper. Here, we also report results in terms of `Best` and `Worst` metrics in Tables 2, 3, 4, 5, where FoMs are computed using `Mean`.

For a model and a given $K$, there are five different lists of ordered top-$K$ targets. For each image, its `Best` (`Worst`) ASR is 1 if any (all) of the five lists of targets can be successfully attacked, and the `Mean` ASR is the fraction of successful attacks over the total five runs. The overall `Best, Mean, Worst` ASRs are then averaged over the 1000 testing images. Corresponding to the three types of ASRs, their $\ell_p$ norms are computed using successfully attacked images only.

# F. More Qualitative Results

We show examples learned by both our RisingAttacK and QuadAttacK for each of the five random seeds. Fig. 3 shows the examples by QuadAttacK, corresponding to those by our RisingAttacK in Fig. 2 and the seed is 42

More examples are in Figs. 4 and 5 (for seed=52).

Due to the file size limit (20M), we will show examples using other seeds in our released code repository.

*Table 2.* Full results including the three metrics (Best, Mean, Worst) for ResNet50 in Table 1(a). FoM is based on the Mean performance.

| Top-$K$ | Method | Best | | | | Mean | | | | Worst | | | | Time (s/img)↓ | FoM↑ |
|---|---|---|---|---|---|---|---|---|---|---|---|---|---|---|---|
| | | ASR↑ | $\ell_1$↓ | $\ell_2$↓ | $\ell_\infty$↓ | ASR↑ | $\ell_1$↓ | $\ell_2$↓ | $\ell_\infty$↓ | ASR↑ | $\ell_1$↓ | $\ell_2$↓ | $\ell_\infty$↓ | | |
| Top-30 | QuadAttacK$_{60}$ | 0.4590 | 11.6907 | 3620.6876 | 0.1307 | 0.2076 | 11.8070 | 3654.9139 | 0.1349 | 0.0330 | 11.9156 | 3686.5432 | 0.1394 | **3.3947** | 6.4793 |
| | RisingAttacK$_{60}$ | **0.8770** | **6.5011** | **1922.6780** | **0.0475** | **0.6642** | **7.0271** | **2081.8960** | **0.0511** | **0.3900** | **7.5992** | **2255.0249** | **0.0550** | 17.0013 | |
| | QuadAttacK$_{30}$ | | | Failed | | | | Failed | | | | Failed | | **1.6539** | inf |
| | RisingAttacK$_{30}$ | **0.0110** | 6.2378 | 1844.3013 | 0.0470 | **0.0022** | 6.2378 | 1844.3013 | 0.0470 | | | Failed | | 8.5619 | |
| Top-25 | QuadAttacK$_{60}$ | 0.8460 | 11.3686 | 3526.3618 | 0.1219 | 0.6018 | 11.6214 | 3599.8101 | 0.1301 | 0.3060 | 11.8774 | 3674.2965 | 0.1388 | **3.4167** | 3.6439 |
| | RisingAttacK$_{60}$ | **0.9580** | **4.8257** | **1419.8702** | **0.0361** | **0.8420** | **5.2960** | **1561.6462** | **0.0393** | **0.6700** | **5.8040** | **1715.5532** | **0.0427** | 14.0839 | |
| | QuadAttacK$_{30}$ | 0.0090 | 10.4263 | 3259.2773 | 0.0991 | 0.0018 | 10.4263 | 3259.2773 | 0.0991 | | | Failed | | **1.7058** | 48.9628 |
| | RisingAttacK$_{30}$ | **0.1270** | **5.0430** | **1487.4868** | **0.0382** | **0.0392** | **5.1218** | **1511.3347** | **0.0388** | **0.0010** | **5.2031** | **1535.9489** | **0.0394** | 7.0999 | |
| Top-20 | QuadAttacK$_{60}$ | **0.9560** | 9.7368 | 3030.3407 | 0.0984 | **0.8344** | 10.0891 | 3133.6199 | 0.1079 | 0.6500 | 10.4514 | 3239.8729 | 0.1183 | **3.4039** | 3.1100 |
| | RisingAttacK$_{60}$ | 0.9500 | **3.3880** | **992.4068** | **0.0257** | 0.8306 | **3.7474** | **1101.1521** | **0.0281** | **0.6520** | **4.1005** | **1207.9307** | **0.0305** | 6.7267 | |
| | QuadAttacK$_{30}$ | **0.2620** | 9.0111 | 2824.4299 | 0.0839 | **0.0978** | 9.0948 | 2850.0433 | 0.0858 | **0.0080** | 9.1756 | 2874.5937 | 0.0876 | **1.7264** | 1.9481 |
| | RisingAttacK$_{30}$ | 0.2040 | **3.4129** | **1000.3753** | **0.0264** | 0.0666 | **3.4854** | **1022.5585** | **0.0269** | 0.0010 | **3.5609** | **1045.7229** | **0.0274** | 3.7216 | |
| Top-15 | QuadAttacK$_{60}$ | 0.9870 | 7.9013 | 2470.1343 | 0.0729 | 0.9440 | 8.3368 | 2600.7510 | 0.0822 | 0.8460 | 8.8010 | 2739.1058 | 0.0932 | **3.4839** | 3.2229 |
| | RisingAttacK$_{60}$ | **0.9990** | **2.6335** | **763.6941** | **0.0208** | **0.9868** | **3.0150** | **878.9222** | **0.0233** | **0.9610** | **3.4246** | **1003.1229** | **0.0260** | 5.1634 | |
| | QuadAttacK$_{30}$ | 0.7540 | 7.5962 | 2380.3027 | 0.0674 | 0.4922 | 7.8296 | 2451.8036 | 0.0717 | 0.2090 | 8.0427 | 2517.2334 | 0.0759 | **1.7382** | 3.3674 |
| | RisingAttacK$_{30}$ | **0.8310** | **2.7910** | **812.1335** | **0.0219** | **0.5856** | **2.9944** | **873.3877** | **0.0234** | **0.2970** | **3.2057** | **936.9361** | **0.0249** | 2.8794 | |
| Top-10 | QuadAttacK$_{60}$ | 0.9970 | 6.1074 | 1917.6238 | 0.0504 | 0.9866 | 6.5228 | 2044.5753 | 0.0576 | 0.9660 | 6.9893 | 2186.1273 | 0.0666 | 3.7396 | 3.3482 |
| | RisingAttacK$_{60}$ | **0.9980** | **1.8077** | **519.1006** | **0.0149** | **0.9936** | **2.0825** | **602.1784** | **0.0167** | **0.9800** | **2.3735** | **690.1587** | **0.0187** | **3.3991** | |
| | QuadAttacK$_{30}$ | **0.9520** | 6.0248 | 1893.9204 | 0.0491 | **0.8460** | 6.3547 | 1994.8023 | 0.0544 | **0.6600** | 6.6744 | 2093.2491 | 0.0598 | **1.7593** | 2.9244 |
| | RisingAttacK$_{30}$ | 0.9410 | **1.9697** | **568.1198** | **0.0160** | 0.8064 | **2.1748** | **630.0922** | **0.0175** | 0.5990 | **2.3743** | **690.3320** | **0.0188** | 1.7965 | |
| Top-5 | QuadAttacK$_{60}$ | **1.0000** | 3.6813 | 1161.6413 | 0.0264 | **0.9968** | 4.0029 | 1261.2314 | 0.0309 | **0.9900** | 4.3529 | 1369.4046 | 0.0362 | 4.5257 | 3.3373 |
| | RisingAttacK$_{60}$ | 0.9890 | **0.9567** | **270.4779** | **0.0085** | 0.9558 | **1.1534** | **330.1495** | **0.0098** | 0.8980 | **1.3547** | **391.1246** | **0.0112** | **1.8225** | |
| | QuadAttacK$_{30}$ | **0.9880** | 3.6540 | 1153.1996 | 0.0261 | **0.9590** | 3.9539 | 1246.4929 | 0.0300 | **0.8950** | 4.2646 | 1343.3064 | 0.0344 | 2.1458 | 2.6681 |
| | RisingAttacK$_{30}$ | 0.9860 | **1.2737** | **361.3176** | **0.0110** | 0.9504 | **1.4693** | **420.0254** | **0.0124** | 0.8910 | **1.6607** | **477.3890** | **0.0138** | **0.9517** | |
| Top-1 | QuadAttacK$_{60}$ | **1.0000** | 1.1548 | 381.4498 | 0.0057 | 0.9996 | 1.4443 | 467.1178 | 0.0083 | 0.9980 | 1.7222 | 550.7556 | 0.0110 | 5.3373 | 2.1564 |
| | RisingAttacK$_{60}$ | **1.0000** | **0.4136** | **104.7782** | **0.0049** | **0.9992** | **0.6144** | **165.8517** | **0.0064** | **0.9990** | **0.8364** | **233.5616** | **0.0079** | **0.6114** | |
| | QuadAttacK$_{30}$ | 0.9980 | 1.1521 | 380.6522 | 0.0057 | 0.9772 | 1.4244 | 461.1199 | 0.0080 | 0.9340 | 1.6861 | 540.0446 | 0.0105 | 2.6411 | 1.4638 |
| | RisingAttacK$_{30}$ | **1.0000** | **0.6218** | **163.8710** | **0.0066** | **0.9986** | **0.9155** | **251.6174** | **0.0088** | **0.9950** | **1.2054** | **338.3424** | **0.0110** | **0.3201** | |

*Table 3.* Full results including the three metrics (Best, Mean, Worst) for DenseNet121 in Table 1(b). FoM is based on the Mean performance.

| Top-$K$ | Method | Best | | | | Mean | | | | Worst | | | | Time (s/img)↓ | FoM↑ |
|---|---|---|---|---|---|---|---|---|---|---|---|---|---|---|---|
| | | ASR↑ | $\ell_1$↓ | $\ell_2$↓ | $\ell_\infty$↓ | ASR↑ | $\ell_1$↓ | $\ell_2$↓ | $\ell_\infty$↓ | ASR↑ | $\ell_1$↓ | $\ell_2$↓ | $\ell_\infty$↓ | | |
| Top-30 | QuadAttacK$_{60}$ | | | Failed | | | | Failed | | | | Failed | | **4.5409** | inf |
| | RisingAttacK$_{60}$ | **0.7490** | 13.8191 | 4117.4537 | 0.0991 | **0.4074** | 14.7263 | 4393.8482 | 0.1051 | **0.0730** | 15.6581 | 4677.7359 | 0.1114 | 20.3156 | |
| | QuadAttacK$_{30}$ | | | Failed | | | | Failed | | | | Failed | | **2.3266** | inf |
| | RisingAttacK$_{30}$ | | | Failed | | | | Failed | | | | Failed | | 10.2335 | |
| Top-25 | QuadAttacK$_{60}$ | 0.4600 | 13.0098 | 4002.5555 | 0.1489 | 0.1734 | 13.1825 | 4053.5759 | 0.1531 | 0.0050 | 13.3529 | 4103.6056 | 0.1573 | **4.1657** | 8.5496 |
| | RisingAttacK$_{60}$ | **0.9860** | **8.8114** | **2589.4709** | **0.0666** | **0.9370** | **9.9898** | **2945.3574** | **0.0747** | **0.8340** | **11.2310** | **3320.9942** | **0.0830** | 16.8643 | |
| | QuadAttacK$_{30}$ | | | Failed | | | | Failed | | | | Failed | | **2.2016** | inf |
| | RisingAttacK$_{30}$ | **0.3150** | 9.6484 | 2840.4239 | 0.0736 | **0.1094** | 9.9203 | 2921.6770 | 0.0756 | **0.0010** | 10.1868 | 3001.6276 | 0.0775 | 8.5279 | |
| Top-20 | QuadAttacK$_{60}$ | 0.9730 | 11.0446 | 3412.9160 | 0.1140 | 0.8340 | 11.6266 | 3583.3589 | 0.1268 | 0.5840 | 12.1967 | 3749.9306 | 0.1403 | **4.0066** | 2.6290 |
| | RisingAttacK$_{60}$ | **0.9970** | **5.1080** | **1480.5618** | **0.0406** | **0.9812** | **5.9921** | **1744.8239** | **0.0468** | **0.9450** | **6.9129** | **2020.8591** | **0.0532** | 8.2901 | |
| | QuadAttacK$_{30}$ | 0.1240 | 9.7962 | 3054.7057 | 0.0914 | 0.0330 | 9.8564 | 3072.6790 | 0.0923 | | | Failed | | **2.0206** | 23.7723 |
| | RisingAttacK$_{30}$ | **0.7320** | **5.7318** | **1664.7979** | **0.0456** | **0.4500** | **6.1377** | **1786.5613** | **0.0485** | **0.1460** | **6.5680** | **1915.5893** | **0.0516** | 4.6070 | |
| Top-15 | QuadAttacK$_{60}$ | 0.9970 | 8.6591 | 2701.7797 | 0.0774 | 0.9866 | 9.2713 | 2884.2755 | 0.0887 | 0.9610 | 9.8890 | 3067.5058 | 0.1011 | **3.8963** | 2.3310 |
| | RisingAttacK$_{60}$ | **1.0000** | **3.8050** | **1086.1460** | **0.0318** | **1.0000** | **4.3657** | **1252.9889** | **0.0359** | **1.0000** | **4.9640** | **1431.3256** | **0.0402** | 6.2878 | |
| | QuadAttacK$_{30}$ | 0.7780 | 8.3044 | 2599.4287 | 0.0720 | 0.5088 | 8.6281 | 2697.9823 | 0.0771 | 0.2380 | 8.9352 | 2791.7948 | 0.0821 | **1.8919** | 3.5524 |
| | RisingAttacK$_{30}$ | **0.9900** | **4.2188** | **1207.4247** | **0.0350** | **0.9362** | **4.7380** | **1362.7350** | **0.0387** | **0.8350** | **5.2677** | **1521.6817** | **0.0425** | 3.5501 | |
| Top-10 | QuadAttacK$_{60}$ | **1.0000** | 6.2172 | 1957.3529 | 0.0469 | 0.9986 | 6.7558 | 2123.4894 | 0.0545 | 0.9970 | 7.3050 | 2291.5986 | 0.0629 | **3.8256** | 2.5458 |
| | RisingAttacK$_{60}$ | **1.0000** | **2.3212** | **649.6995** | **0.0208** | **1.0000** | **2.6903** | **759.1986** | **0.0235** | **1.0000** | **3.0749** | **874.1637** | **0.0263** | 4.2223 | |
| | QuadAttacK$_{30}$ | 0.9900 | 6.1801 | 1946.4525 | 0.0464 | 0.9392 | 6.6701 | 2098.0095 | 0.0531 | 0.8380 | 7.1563 | 2248.2525 | 0.0602 | **1.8918** | 2.4272 |
| | RisingAttacK$_{30}$ | **0.9980** | **2.4880** | **698.6799** | **0.0221** | **0.9880** | **2.9210** | **827.1606** | **0.0253** | **0.9690** | **3.3607** | **958.0512** | **0.0285** | 2.2937 | |
| Top-5 | QuadAttacK$_{60}$ | **1.0000** | 3.6045 | 1143.1525 | 0.0226 | 0.9998 | 3.9671 | 1258.1706 | 0.0264 | 0.9990 | 4.3422 | 1377.3171 | 0.0305 | 3.8644 | 3.0870 |
| | RisingAttacK$_{60}$ | **1.0000** | **1.0122** | **270.9332** | **0.0103** | **0.9994** | **1.2169** | **331.6714** | **0.0119** | **0.9980** | **1.4422** | **398.9840** | **0.0136** | 2.2643 | |
| | QuadAttacK$_{30}$ | 0.9980 | 3.6019 | 1142.1894 | 0.0226 | 0.9924 | 3.9526 | 1253.5745 | 0.0262 | 0.9820 | 4.3229 | 1371.6348 | 0.0303 | 1.8502 | 2.2794 |
| | RisingAttacK$_{30}$ | **1.0000** | **1.3896** | **378.3550** | **0.0135** | **0.9982** | **1.6603** | **457.9204** | **0.0156** | **0.9940** | **1.9524** | **544.3561** | **0.0178** | **1.2082** | |
| Top-1 | QuadAttacK$_{60}$ | **1.0000** | 1.1724 | 397.8131 | 0.0049 | **1.0000** | 1.5191 | 503.0047 | 0.0070 | **1.0000** | 1.8544 | 606.7211 | 0.0093 | 3.0413 | 1.9466 |
| | RisingAttacK$_{60}$ | **1.0000** | **0.4732** | **112.4793** | **0.0063** | **1.0000** | **0.7001** | **177.1356** | **0.0085** | **1.0000** | **0.9566** | **251.8730** | **0.0108** | 0.8046 | |
| | QuadAttacK$_{30}$ | **1.0000** | 1.1728 | 397.9329 | 0.0049 | 0.9960 | 1.5144 | 501.4779 | 0.0070 | 0.9880 | 1.8426 | 603.0238 | 0.0092 | 1.5519 | 1.2739 |
| | RisingAttacK$_{30}$ | **1.0000** | **0.7032** | **175.2027** | **0.0084** | **1.0000** | **1.0708** | **280.0300** | **0.0116** | **1.0000** | **1.4523** | **390.3135** | **0.0149** | **0.4255** | |

Table 4. Full results including the three metrics (Best, Mean, Worst) for ViT-B in Table 1(c). FoM is based on the Mean performance.

| Top-K | Method | Best ASR↑ | ℓ1↓ | ℓ2↓ | ℓ∞↓ | Mean ASR↑ | ℓ1↓ | ℓ2↓ | ℓ∞↓ | Worst ASR↑ | ℓ1↓ | ℓ2↓ | ℓ∞↓ | Time (s/img)↓ | FoM↑ |
|---|---|---|---|---|---|---|---|---|---|---|---|---|---|---|---|
| Top-30 | QuadAttacK60 | 0.6720 | 9.4223 | 2864.1279 | 0.0996 | 0.3272 | **9.6708** | 2938.2587 | 0.1032 | 0.0590 | **9.9254** | **3014.5189** | **0.1067** | 5.2135 | 3.1589 |
|  | RisingAttacK60 | **1.0000** | **6.7803** | **1881.7456** | **0.0613** | **0.9534** | 9.7262 | **2721.2876** | **0.0876** | **0.8260** | 14.2641 | 4001.9670 | 0.1299 | 43.3954 | |
|  | QuadAttacK30 | Failed | | | | Failed | | | | Failed | | | | 2.7870 | inf |
|  | RisingAttacK30 | **0.8080** | 10.3031 | 2880.8575 | 0.0934 | **0.5568** | 11.4132 | 3206.4565 | 0.1029 | **0.2430** | 12.6194 | 3558.5805 | 0.1134 | 21.7179 | |
| Top-25 | QuadAttacK60 | 0.9350 | 9.0105 | 2737.0924 | 0.0935 | 0.6872 | 9.4331 | 2860.6667 | 0.1002 | 0.3490 | 9.8582 | 2985.2203 | 0.1070 | 5.2723 | 2.6425 |
|  | RisingAttacK60 | **1.0000** | **4.0101** | **1084.9150** | **0.0382** | **0.9944** | **5.5706** | **1520.5703** | **0.0526** | **0.9740** | **8.1685** | **2232.4551** | **0.0782** | 36.0486 | |
|  | QuadAttacK30 | Failed | | | | Failed | | | | Failed | | | | 2.7354 | inf |
|  | RisingAttacK30 | **0.9330** | 6.8418 | 1877.8280 | 0.0643 | **0.7536** | 7.7050 | 2126.3211 | 0.0721 | **0.5000** | 8.8041 | 2438.7838 | 0.0825 | 18.0775 | |
| Top-20 | QuadAttacK60 | 0.9710 | 7.4811 | 2268.2729 | 0.0748 | 0.7828 | 7.9108 | 2393.0875 | 0.0815 | 0.4910 | 8.3426 | 2519.7231 | 0.0884 | 5.0069 | 2.8308 |
|  | RisingAttacK60 | **0.9980** | **3.0033** | **792.6934** | 0.0294 | **0.9864** | **3.7609** | **1007.6887** | **0.0360** | **0.9560** | **4.5804** | **1241.9193** | **0.0432** | 15.8230 | |
|  | QuadAttacK30 | 0.0020 | 6.3770 | 1992.7502 | 0.0533 | 0.0004 | 6.3770 | 1992.7502 | 0.0533 | Failed | | | | 2.6210 | 1610.8632 |
|  | RisingAttacK30 | **0.7380** | **4.5276** | **1221.1645** | **0.0436** | **0.4956** | **4.9482** | **1343.2135** | **0.0473** | **0.2490** | **5.3892** | **1471.1873** | **0.0511** | 7.9615 | |
| Top-15 | QuadAttacK60 | 0.9730 | 5.8803 | 1780.9271 | 0.0558 | 0.8404 | 6.2661 | 1893.5173 | 0.0620 | 0.5980 | 6.6687 | 2011.4965 | 0.0684 | 4.7622 | 2.7231 |
|  | RisingAttacK60 | **1.0000** | **2.3069** | **593.4747** | 0.0235 | **0.9988** | **2.8751** | **753.1852** | 0.0284 | **0.9940** | **3.5373** | **939.6227** | **0.0343** | 11.9841 | |
|  | QuadAttacK30 | 0.0240 | 4.7795 | 1490.0706 | 0.0382 | 0.0056 | 4.7982 | 1495.8188 | 0.0385 | Failed | | | | 2.4245 | 164.8583 |
|  | RisingAttacK30 | **0.9270** | **3.4401** | **908.2324** | **0.0339** | **0.7510** | **3.8944** | **1038.7394** | **0.0379** | **0.5310** | **4.4471** | **1197.6996** | **0.0427** | 6.0305 | |
| Top-10 | QuadAttacK60 | 0.9900 | 4.1855 | 1274.5181 | 0.0363 | 0.9130 | 4.5246 | 1374.2282 | 0.0410 | 0.7510 | 4.8729 | 1476.9954 | 0.0459 | 4.6368 | 2.5247 |
|  | RisingAttacK60 | **1.0000** | **1.5872** | **397.2939** | 0.0169 | **0.9936** | **1.9915** | **508.8791** | 0.0206 | **0.9750** | **2.4423** | **634.1043** | 0.0247 | 8.2583 | |
|  | QuadAttacK30 | 0.0810 | 3.4468 | 1078.1791 | 0.0256 | 0.0252 | 3.4999 | 1094.6987 | 0.0261 | Failed | | | | 2.3034 | 36.7947 |
|  | RisingAttacK30 | **0.9010** | **2.3233** | **599.0848** | 0.0240 | **0.7112** | **2.6247** | **684.1576** | **0.0267** | **0.4810** | **2.9492** | **775.8982** | 0.0297 | 4.1602 | |
| Top-5 | QuadAttacK60 | **1.0000** | 3.2461 | 1010.5003 | 0.0241 | **0.9980** | 3.6439 | 1128.3054 | 0.0288 | **0.9930** | 4.0423 | 1246.0157 | 0.0338 | 4.3981 | 1.7630 |
|  | RisingAttacK60 | 0.8280 | **0.9934** | **245.3329** | 0.0111 | 0.5712 | **1.1650** | **292.6494** | 0.0128 | 0.2910 | **1.3395** | **340.9352** | 0.0144 | 4.4038 | |
|  | QuadAttacK30 | 0.8120 | 3.0924 | 968.7317 | 0.0221 | 0.5024 | 3.2930 | 1029.8490 | 0.0242 | 0.1780 | 3.4820 | 1087.7387 | 0.0261 | 2.1108 | 2.3688 |
|  | RisingAttacK30 | **0.8420** | **1.3886** | **345.5126** | **0.0153** | **0.5980** | **1.6101** | **406.4644** | **0.0174** | **0.3310** | **1.8357** | **468.6344** | **0.0195** | 2.2197 | |
| Top-1 | QuadAttacK60 | **1.0000** | 1.2537 | 410.8221 | 0.0059 | **0.9998** | 1.5736 | 509.7575 | 0.0081 | **0.9990** | 1.9042 | 612.7661 | 0.0106 | 2.6007 | 3.2121 |
|  | RisingAttacK60 | 0.9940 | **0.2978** | **60.3734** | **0.0045** | 0.9388 | **0.4365** | **96.0745** | **0.0060** | 0.8260 | **0.6048** | **139.9744** | **0.0078** | 1.2715 | |
|  | QuadAttacK30 | **1.0000** | 1.2541 | 410.9606 | 0.0059 | **0.9958** | 1.5681 | 508.0591 | 0.0081 | **0.9900** | 1.8935 | 609.5179 | 0.0105 | 1.3040 | 2.1102 |
|  | RisingAttacK30 | 0.9950 | **0.4270** | **90.9070** | 0.0060 | 0.9362 | **0.6578** | **149.1661** | 0.0086 | 0.8180 | **0.9318** | **219.6820** | 0.0115 | 0.6417 | |

Table 5. Full results including the three metrics (Best, Mean, Worst) for DEiT-B in Table 1(d). FoM is based on the Mean performance.

| Top-K | Method | Best ASR↑ | ℓ1↓ | ℓ2↓ | ℓ∞↓ | Mean ASR↑ | ℓ1↓ | ℓ2↓ | ℓ∞↓ | Worst ASR↑ | ℓ1↓ | ℓ2↓ | ℓ∞↓ | Time (s/img)↓ | FoM↑ |
|---|---|---|---|---|---|---|---|---|---|---|---|---|---|---|---|
| Top-30 | QuadAttacK60 | 0.2350 | 9.3006 | 2840.2109 | 0.0985 | 0.0640 | **9.3734** | 2860.9240 | 0.0997 | 0.0010 | **9.4465** | **2881.7991** | 0.1009 | **4.1792** | 8.8333 |
|  | RisingAttacK60 | **0.9860** | **7.9531** | **2263.3389** | **0.0678** | **0.5150** | 9.4432 | **2697.9176** | **0.0804** | **0.0340** | 11.7557 | 3365.5665 | **0.1007** | 43.3521 | |
|  | QuadAttacK30 | Failed | | | | Failed | | | | Failed | | | | 2.3032 | inf |
|  | RisingAttacK30 | **0.2160** | 10.8323 | 3092.9352 | 0.0937 | **0.0600** | 11.0771 | 3165.6910 | 0.0957 | Failed | | | | 21.6930 | |
| Top-25 | QuadAttacK60 | 0.9900 | 8.8805 | 2703.0149 | 0.0886 | 0.8644 | 9.3780 | 2849.8222 | 0.0960 | 0.5880 | 9.9093 | 3006.5687 | 0.1039 | **4.0966** | 2.1975 |
|  | RisingAttacK60 | **1.0000** | **3.6473** | **1002.9836** | **0.0337** | **0.9854** | **5.1921** | **1434.6160** | **0.0482** | **0.9360** | **7.9650** | **2187.8969** | **0.0768** | 36.1084 | |
|  | QuadAttacK30 | Failed | | | | Failed | | | | Failed | | | | 2.2173 | inf |
|  | RisingAttacK30 | **0.9100** | 5.6674 | 1575.9766 | 0.0520 | **0.6748** | 6.3220 | 1763.4334 | 0.0581 | **0.3620** | 7.1691 | 2002.6759 | 0.0662 | 18.1108 | |
| Top-20 | QuadAttacK60 | 0.9990 | 7.2048 | 2199.2260 | 0.0665 | 0.9612 | 7.6974 | 2343.5441 | 0.0735 | 0.8540 | 8.2263 | 2499.1404 | 0.0811 | **4.1868** | 2.8466 |
|  | RisingAttacK60 | **1.0000** | **2.3202** | **611.1048** | 0.0230 | **0.9956** | **2.9174** | **781.2607** | 0.0282 | **0.9850** | **3.5733** | **968.9946** | **0.0339** | 15.8331 | |
|  | QuadAttacK30 | 0.0160 | 6.2491 | 1950.0525 | 0.0524 | 0.0032 | 6.2491 | 1950.0525 | 0.0524 | Failed | | | | 2.1503 | 325.7059 |
|  | RisingAttacK30 | **0.8560** | **3.5031** | **946.6329** | **0.0338** | **0.6348** | **3.8373** | **1045.3953** | **0.0366** | **0.3810** | **4.1863** | **1148.5782** | 0.0395 | 7.9624 | |
| Top-15 | QuadAttacK60 | **1.0000** | 5.5928 | 1713.0918 | 0.0481 | 0.9750 | 6.0671 | 1852.4958 | 0.0544 | 0.9100 | 6.5599 | 1997.5902 | 0.0611 | **3.9525** | 2.8819 |
|  | RisingAttacK60 | 1.0000 | **1.7594** | **448.6124** | 0.0184 | **1.0000** | **2.2015** | **573.3811** | 0.0223 | **1.0000** | **2.7017** | **715.2705** | 0.0266 | 11.9810 | |
|  | QuadAttacK30 | 0.1350 | 4.9541 | 1548.1232 | 0.0383 | 0.0338 | 4.9874 | 1561.1460 | 0.0386 | Failed | | | | 2.0234 | 42.8983 |
|  | RisingAttacK30 | **0.9870** | **2.7227** | **714.8364** | 0.0273 | **0.9278** | **3.1490** | **838.2295** | 0.0310 | **0.8140** | **3.5773** | **963.3399** | 0.0346 | 6.0263 | |
| Top-10 | QuadAttacK60 | 0.9980 | 3.9545 | 1222.8665 | 0.0305 | 0.9762 | 4.3693 | 1346.6326 | 0.0353 | 0.9150 | 4.7999 | 1475.4252 | 0.0406 | **3.8755** | 2.9455 |
|  | RisingAttacK60 | **1.0000** | **1.1885** | **291.5197** | 0.0132 | **0.9996** | **1.5076** | **379.6582** | 0.0162 | **0.9980** | **1.8825** | **484.4438** | 0.0195 | 8.2610 | |
|  | QuadAttacK30 | 0.3540 | 3.4965 | 1097.7370 | 0.0249 | 0.1298 | 3.5782 | 1123.3760 | 0.0256 | 0.0100 | 3.6597 | 1148.9145 | 0.0263 | 1.9552 | 11.4300 |
|  | RisingAttacK30 | **0.9800** | **1.7993** | **458.0865** | **0.0191** | **0.9200** | **2.1465** | **556.2741** | **0.0222** | **0.8000** | **2.5030** | **658.8665** | **0.0253** | 4.1613 | |
| Top-5 | QuadAttacK60 | **1.0000** | 2.9872 | 940.1374 | 0.0201 | 0.9984 | 3.3975 | 1064.7252 | 0.0243 | 0.9950 | 3.8203 | 1192.7754 | 0.0288 | **3.4381** | 3.1378 |
|  | RisingAttacK60 | 1.0000 | **0.8108** | **189.0044** | **0.0096** | **0.9992** | **1.0575** | **254.5953** | **0.0121** | **0.9970** | **1.3365** | **329.6166** | **0.0148** | 4.4027 | |
|  | QuadAttacK30 | 0.9700 | 2.9571 | 932.7132 | 0.0197 | 0.7794 | 3.2526 | 1024.0607 | 0.0225 | 0.4770 | 3.5452 | 1114.4266 | 0.0252 | **1.7718** | 2.6286 |
|  | RisingAttacK30 | **0.9760** | **1.0890** | **264.4306** | **0.0125** | **0.8800** | **1.3450** | **334.9398** | **0.0149** | **0.7110** | **1.6168** | **410.8617** | **0.0174** | 2.2165 | |
| Top-1 | QuadAttacK60 | **1.0000** | 1.1376 | 381.0736 | 0.0047 | **1.0000** | 1.3910 | 459.6084 | 0.0063 | **1.0000** | 1.6659 | 545.2655 | 0.0080 | 2.9955 | 3.9437 |
|  | RisingAttacK60 | 0.9990 | **0.2450** | **46.1936** | **0.0041** | 0.9794 | **0.3340** | **68.5738** | **0.0052** | 0.9420 | **0.4472** | **97.7384** | **0.0065** | **1.2708** | |
|  | QuadAttacK30 | **1.0000** | 1.1372 | 380.9527 | 0.0047 | **0.9994** | 1.3899 | 459.3060 | 0.0063 | **0.9970** | 1.6631 | 544.4684 | 0.0080 | 1.4404 | 2.4502 |
|  | RisingAttacK30 | 1.0000 | **0.3367** | **67.5065** | 0.0052 | 0.9772 | **0.5249** | **114.2980** | 0.0073 | 0.9240 | **0.7524** | **171.9542** | 0.0099 | 0.6426 | |

*Table 6.* **Ordered top-$K$ attack results using the lowest-K predictions of benign images as attack targets**. Overall, our RisingAttacK shows a big leap forward in advancing ordered top-$K$ attacks, outperforming the prior state-of-the-art method, QuadAttacK (Paniagua et al., 2023) by a large margin in most cases (higher ASRs with lower $\ell_p$ norms). $\ell_\infty$-norms in red is to show they are treated as being "visually imperceptible" based on the commonly used threshold $8/255 = 0.0314$. The subscripts of methods (30 and 60) represent the computing budgets. The FoM (figure of merits) of our RisingAttacK against QuadAttacK is computed by Eqn. 23 to show its holistic improvement in terms of how many times it is better.

(a) ResNet-50 (He et al., 2016)

| Top-$K$ | Method | Single-Run | | | | Time (s/img) ↓ | FoM↑ |
|---|---|---|---|---|---|---|---|
| | | ASR ↑ | $\ell_1$ ↓ | $\ell_2$ ↓ | $\ell_\infty$ ↓ | | |
| Top-30 | QuadAttacK$_{60}$ | 0.3250 | 11.7308 | 3640.1040 | 0.1292 | **4.7020** | 3.8464 |
| | RisingAttacK$_{60}$ | **0.5340** | **5.9437** | **1762.7836** | **0.0433** | 38.9733 | |
| | QuadAttacK$_{30}$ | 0.0010 | 10.6442 | 3329.8271 | 0.1111 | **2.2976** | 7.6272 |
| | RisingAttacK$_{30}$ | **0.0040** | **6.3581** | **1870.2285** | **0.0490** | 19.6176 | |
| Top-25 | QuadAttacK$_{60}$ | 0.6240 | 11.5386 | 3585.0978 | 0.1250 | **4.5973** | 3.1978 |
| | RisingAttacK$_{60}$ | **0.7070** | **4.7898** | **1415.3054** | **0.0355** | 32.0203 | |
| | QuadAttacK$_{30}$ | **0.0320** | 10.3092 | 3233.3005 | 0.0954 | **2.2886** | 2.2899 |
| | RisingAttacK$_{30}$ | 0.0280 | **4.2926** | **1263.4319** | **0.0330** | 16.1438 | |
| Top-20 | QuadAttacK$_{60}$ | **0.8090** | 10.0532 | 3129.2405 | 0.1050 | **4.5393** | 2.8243 |
| | RisingAttacK$_{60}$ | 0.7300 | **3.6953** | **1088.5812** | **0.0277** | 14.1311 | |
| | QuadAttacK$_{30}$ | **0.1270** | 9.1247 | 2863.4929 | 0.0830 | **2.2579** | 1.2784 |
| | RisingAttacK$_{30}$ | 0.0560 | **3.3960** | **993.0273** | **0.0265** | 7.4246 | |
| Top-15 | QuadAttacK$_{60}$ | 0.9370 | 8.4982 | 2653.1828 | 0.0840 | **4.5458** | 3.1157 |
| | RisingAttacK$_{60}$ | **0.9620** | **3.1353** | **917.7340** | **0.0240** | 10.7638 | |
| | QuadAttacK$_{30}$ | **0.3980** | 7.8166 | 2453.6085 | 0.0696 | **2.2794** | 2.6514 |
| | RisingAttacK$_{30}$ | 0.3880 | **3.0805** | **900.1031** | **0.0240** | 5.6529 | |
| Top-10 | QuadAttacK$_{60}$ | 0.9840 | 6.7946 | 2130.0686 | 0.0610 | **4.6965** | 3.1959 |
| | RisingAttacK$_{60}$ | 0.9840 | **2.2788** | **661.9469** | **0.0180** | 7.6067 | |
| | QuadAttacK$_{30}$ | **0.7670** | 6.4958 | 2040.6640 | 0.0554 | **2.3425** | 2.5281 |
| | RisingAttacK$_{30}$ | 0.6590 | **2.3197** | **674.2584** | **0.0185** | 3.9934 | |
| Top-5 | QuadAttacK$_{60}$ | **0.9910** | 4.2898 | 1351.5930 | 0.0339 | 5.1849 | 2.9681 |
| | RisingAttacK$_{60}$ | 0.9290 | **1.3772** | **397.4290** | **0.0114** | **4.3419** | |
| | QuadAttacK$_{30}$ | **0.9210** | 4.1759 | 1316.8696 | 0.0323 | 2.5234 | 2.4573 |
| | RisingAttacK$_{30}$ | 0.9070 | **1.6905** | **486.2660** | **0.0140** | **2.2706** | |
| Top-1 | QuadAttacK$_{60}$ | 0.9990 | 1.6902 | 542.3103 | 0.0103 | 6.1673 | 2.1155 |
| | RisingAttacK$_{60}$ | **1.0000** | **0.7436** | **203.6175** | **0.0074** | **1.6483** | |
| | QuadAttacK$_{30}$ | 0.9620 | 1.6317 | 523.8288 | 0.0097 | 3.0124 | 1.4446 |
| | RisingAttacK$_{30}$ | **1.0000** | **1.0940** | **303.8991** | 0.0102 | **0.8588** | |

(b) DenseNet-121 (Huang et al., 2017)

| Top-$K$ | Method | Single-Run | | | | Time (s/img) ↓ | FoM↑ |
|---|---|---|---|---|---|---|---|
| | | ASR ↑ | $\ell_1$ ↓ | $\ell_2$ ↓ | $\ell_\infty$ ↓ | | |
| Top-30 | QuadAttacK$_{60}$ | 0.1370 | 13.3362 | 4119.1202 | 0.1466 | **5.3182** | 9.4906 |
| | RisingAttacK$_{60}$ | **0.8520** | **10.2572** | **3029.0433** | **0.0764** | 40.2688 | |
| | QuadAttacK$_{30}$ | Failed | | | | **2.8216** | inf |
| | RisingAttacK$_{30}$ | **0.0850** | 8.7632 | 2561.6755 | 0.0683 | 20.1358 | |
| Top-25 | QuadAttacK$_{60}$ | 0.6760 | 13.1121 | 4052.3545 | 0.1417 | **5.0749** | 3.0375 |
| | RisingAttacK$_{60}$ | **0.9840** | **7.2225** | **2110.8964** | **0.0561** | 32.7520 | |
| | QuadAttacK$_{30}$ | 0.0080 | 11.1429 | 3480.9835 | 0.1047 | **2.6689** | 94.7409 |
| | RisingAttacK$_{30}$ | **0.4730** | **7.5557** | **2203.2915** | **0.0597** | 16.4573 | |
| Top-20 | QuadAttacK$_{60}$ | 0.9360 | 11.3239 | 3513.0089 | 0.1142 | **4.8813** | 2.6289 |
| | RisingAttacK$_{60}$ | **0.9900** | **5.0476** | **1460.8408** | **0.0407** | 14.5282 | |
| | QuadAttacK$_{30}$ | 0.1710 | 9.9557 | 3118.5042 | 0.0886 | **2.5958** | 7.5775 |
| | RisingAttacK$_{30}$ | **0.6840** | 5.5711 | 1612.6615 | 0.0451 | 7.4920 | |
| Top-15 | QuadAttacK$_{60}$ | 0.9910 | 9.2727 | 2898.0552 | 0.0840 | **4.8481** | 2.3800 |
| | RisingAttacK$_{60}$ | **1.0000** | **4.1631** | **1190.6402** | **0.0348** | 11.1324 | |
| | QuadAttacK$_{30}$ | 0.6320 | 8.6222 | 2707.6770 | 0.0731 | **2.4824** | 2.9579 |
| | RisingAttacK$_{30}$ | **0.9540** | **4.5707** | **1310.9248** | **0.0380** | 5.7205 | |
| Top-10 | QuadAttacK$_{60}$ | **0.9980** | 7.0847 | 2232.9939 | 0.0559 | **4.7497** | 2.4674 |
| | RisingAttacK$_{60}$ | **1.0000** | **2.8950** | **817.3569** | **0.0253** | 7.9408 | |
| | QuadAttacK$_{30}$ | 0.9230 | 6.8696 | 2167.0029 | 0.0531 | **2.4163** | 2.3308 |
| | RisingAttacK$_{30}$ | **0.9830** | **3.1446** | **890.9174** | **0.0272** | 4.0917 | |
| Top-5 | QuadAttacK$_{60}$ | **0.9990** | 4.3886 | 1396.2655 | **0.0290** | 4.4818 | 2.8989 |
| | RisingAttacK$_{60}$ | 0.9980 | **1.4362** | **394.3187** | **0.0138** | 4.6659 | |
| | QuadAttacK$_{30}$ | 0.9810 | 4.3306 | 1377.5596 | **0.0284** | 2.2109 | 2.0768 |
| | RisingAttacK$_{30}$ | **0.9930** | **2.0112** | **558.1958** | **0.0185** | 2.3818 | |
| Top-1 | QuadAttacK$_{60}$ | **1.0000** | 1.7865 | 587.0397 | **0.0086** | 3.6998 | 2.0325 |
| | RisingAttacK$_{60}$ | **1.0000** | **0.7916** | **201.4099** | 0.0093 | 2.0500 | |
| | QuadAttacK$_{30}$ | 0.9910 | 1.7586 | 577.6225 | **0.0084** | 1.8592 | 1.3071 |
| | RisingAttacK$_{30}$ | **1.0000** | **1.2213** | **321.1319** | 0.0130 | 1.0587 | |

(c) ViT-B (Dosovitskiy et al., 2020)

| Top-$K$ | Method | Single-Run | | | | Time (s/img) ↓ | FoM↑ |
|---|---|---|---|---|---|---|---|
| | | ASR ↑ | $\ell_1$ ↓ | $\ell_2$ ↓ | $\ell_\infty$ ↓ | | |
| Top-30 | QuadAttacK$_{60}$ | 0.2400 | 9.2933 | 2828.3158 | 0.0997 | **6.5384** | 6.0806 |
| | RisingAttacK$_{60}$ | **0.9980** | **6.9512** | **1923.6753** | **0.0631** | 107.4773 | |
| | QuadAttacK$_{30}$ | Failed | | | | **3.6705** | inf |
| | RisingAttacK$_{30}$ | **0.5200** | 9.2443 | 2576.2629 | 0.0842 | 54.5843 | |
| Top-25 | QuadAttacK$_{60}$ | 0.5220 | 9.2439 | 2813.6584 | 0.0980 | **6.3901** | 3.9297 |
| | RisingAttacK$_{60}$ | **0.9960** | **4.8564** | **1318.3544** | **0.0458** | 90.1623 | |
| | QuadAttacK$_{30}$ | Failed | | | | **3.5624** | inf |
| | RisingAttacK$_{30}$ | **0.5860** | 6.8079 | 1869.8754 | 0.0640 | 46.2905 | |
| Top-20 | QuadAttacK$_{60}$ | 0.6870 | 7.7073 | 2337.6029 | 0.0803 | **6.1960** | 3.2785 |
| | RisingAttacK$_{60}$ | **0.9680** | **3.5511** | **945.7173** | **0.0343** | 43.4669 | |
| | QuadAttacK$_{30}$ | 0.0010 | 5.6397 | 1745.0922 | 0.0501 | | 450.2016 |
| | RisingAttacK$_{30}$ | **0.3810** | **4.8757** | **1316.7148** | **0.0472** | 24.6167 | |
| Top-15 | QuadAttacK$_{60}$ | 0.7880 | 6.0980 | 1849.0582 | 0.0606 | **6.0093** | 2.8951 |
| | RisingAttacK$_{60}$ | **1.0000** | **2.8185** | **734.7598** | **0.0280** | 32.9165 | |
| | QuadAttacK$_{30}$ | 0.0130 | 4.5583 | 1422.9767 | **0.0378** | **3.2740** | 62.7339 |
| | RisingAttacK$_{30}$ | **0.6740** | **3.7918** | **1009.0854** | **0.0372** | 18.6537 | |
| Top-10 | QuadAttacK$_{60}$ | 0.8900 | 4.4073 | 1342.7171 | 0.0401 | 5.7166 | 2.5217 |
| | RisingAttacK$_{60}$ | **0.9940** | **1.9987** | **508.1560** | **0.0208** | 22.5797 | |
| | QuadAttacK$_{30}$ | 0.0310 | 3.2038 | 1007.5255 | **0.0245** | 3.0683 | 24.9115 |
| | RisingAttacK$_{30}$ | **0.6540** | **2.6736** | **696.4219** | 0.0273 | 12.7669 | |
| Top-5 | QuadAttacK$_{60}$ | **0.9970** | 3.6834 | 1140.4541 | **0.0296** | 5.3490 | 1.6908 |
| | RisingAttacK$_{60}$ | 0.5460 | **1.1798** | **295.8843** | **0.0129** | 12.1849 | |
| | QuadAttacK$_{30}$ | 0.4230 | 3.1510 | 987.3517 | **0.0235** | 2.8011 | 2.3368 |
| | RisingAttacK$_{30}$ | **0.5450** | **1.6973** | **429.3175** | **0.0183** | 6.8619 | |
| Top-1 | QuadAttacK$_{60}$ | **1.0000** | 1.7206 | 555.2969 | **0.0091** | **3.4851** | 3.0815 |
| | RisingAttacK$_{60}$ | 0.9260 | **0.4883** | **109.9290** | **0.0065** | 3.7639 | |
| | QuadAttacK$_{30}$ | **0.9970** | 1.7137 | 553.1291 | **0.0091** | **1.7672** | 2.0013 |
| | RisingAttacK$_{30}$ | 0.9250 | **0.7459** | **172.3669** | **0.0094** | 2.0927 | |

(d) DEiT-B (Touvron et al., 2021)

| Top-$K$ | Method | Single-Run | | | | Time (s/img) ↓ | FoM↑ |
|---|---|---|---|---|---|---|---|
| | | ASR ↑ | $\ell_1$ ↓ | $\ell_2$ ↓ | $\ell_\infty$ ↓ | | |
| Top-30 | QuadAttacK$_{60}$ | 0.3660 | 9.6171 | 2927.7147 | 0.0983 | **5.3128** | 4.2723 |
| | RisingAttacK$_{60}$ | **0.9850** | **6.5547** | **1846.3443** | **0.0575** | 107.3817 | |
| | QuadAttacK$_{30}$ | Failed | | | | **3.0631** | inf |
| | RisingAttacK$_{30}$ | **0.2920** | 7.8282 | 2213.4546 | 0.0694 | 54.7059 | |
| Top-25 | QuadAttacK$_{60}$ | 0.8810 | 9.3748 | 2858.4162 | 0.0933 | **5.2609** | 2.9615 |
| | RisingAttacK$_{60}$ | **0.9980** | **3.7927** | **1035.3009** | **0.0357** | 90.7822 | |
| | QuadAttacK$_{30}$ | Failed | | | | **2.9321** | inf |
| | RisingAttacK$_{30}$ | **0.7090** | 5.4817 | 1521.3469 | 0.0508 | 46.5786 | |
| Top-20 | QuadAttacK$_{60}$ | 0.9480 | 7.7045 | 2353.1867 | 0.0729 | **5.2085** | 3.1254 |
| | RisingAttacK$_{60}$ | **0.9910** | **2.6716** | **707.9259** | **0.0264** | 43.5567 | |
| | QuadAttacK$_{30}$ | 0.0040 | 6.1241 | 1920.0202 | 0.0490 | **2.8225** | 246.9753 |
| | RisingAttacK$_{30}$ | **0.5990** | **3.6926** | **1001.5668** | **0.0357** | 24.6677 | |
| Top-15 | QuadAttacK$_{60}$ | 0.9710 | 6.0495 | 1855.0592 | 0.0536 | **4.9720** | 3.0621 |
| | RisingAttacK$_{60}$ | **1.0000** | **2.0686** | **532.7703** | **0.0213** | 32.9756 | |
| | QuadAttacK$_{30}$ | 0.0240 | 4.7361 | 1492.8217 | 0.0364 | **2.7253** | 60.2646 |
| | RisingAttacK$_{30}$ | **0.9050** | **2.9258** | **774.8312** | **0.0291** | 18.6605 | |
| Top-10 | QuadAttacK$_{60}$ | 0.9830 | 4.3244 | 1340.7801 | 0.0344 | **4.7669** | 3.0775 |
| | RisingAttacK$_{60}$ | **0.9990** | **1.4164** | **351.9301** | **0.0155** | 22.6131 | |
| | QuadAttacK$_{30}$ | 0.0940 | 3.3065 | 1037.0081 | **0.0246** | 2.6114 | 15.6515 |
| | RisingAttacK$_{30}$ | **0.9220** | **2.0227** | **520.4555** | 0.0212 | 12.7832 | |
| Top-5 | QuadAttacK$_{60}$ | 0.9980 | 3.4488 | 1085.9038 | **0.0240** | 4.5690 | 3.3363 |
| | RisingAttacK$_{60}$ | **0.9990** | **1.0086** | **240.3898** | **0.0117** | 12.1753 | |
| | QuadAttacK$_{30}$ | 0.7800 | 3.2362 | 1023.7461 | **0.0219** | 2.3610 | 2.6224 |
| | RisingAttacK$_{30}$ | **0.8700** | **1.3222** | **328.4794** | **0.0147** | 6.8720 | |
| Top-1 | QuadAttacK$_{60}$ | **1.0000** | 1.5513 | 509.9606 | **0.0071** | 3.8654 | 4.0412 |
| | RisingAttacK$_{60}$ | 0.9860 | **0.3632** | **76.0298** | **0.0054** | 3.7549 | |
| | QuadAttacK$_{30}$ | **0.9970** | 1.5428 | 507.5080 | **0.0070** | **1.9139** | 2.4961 |
| | RisingAttacK$_{30}$ | 0.9800 | **0.5707** | **126.7541** | **0.0077** | 2.0951 | |

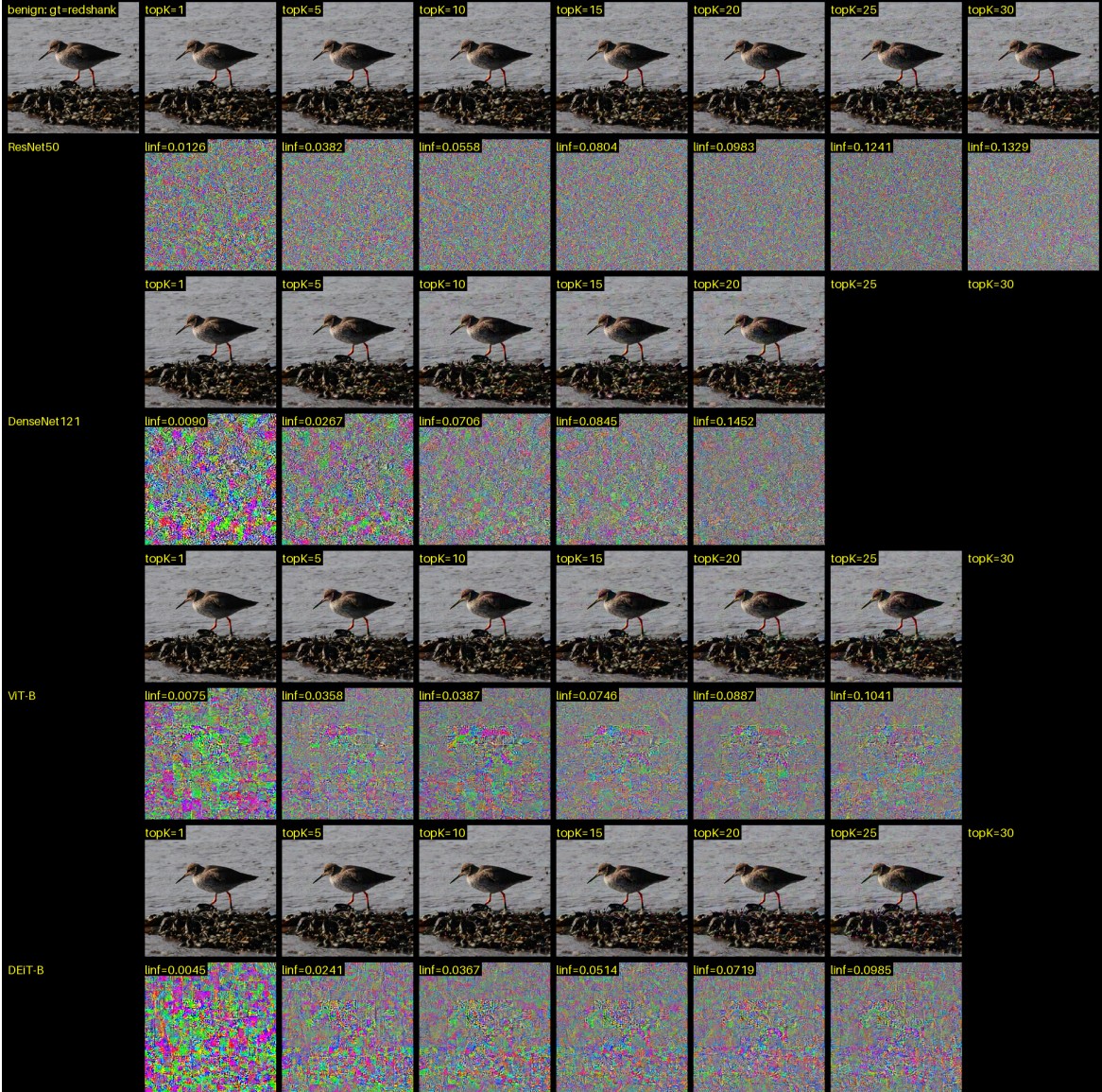

*Figure 3.* **QuadAttacK** examples of adversarial examples and associated perturbations learned for a benign image (ILSVRC2012_val_00002633 with the ground-truth label, `redshank`) using a list of randomly sampled 30 targets (see the list for **seed=42** in the Appendix D) in the order of: `mask`, `analog-clock`, `slide-rule`, `Siberian-husky`, `harmonica`, `African-chameleon`, `dowitcher`, `hyena`, `wing`, `pillow`, `garter-snake`, `Great-Pyrenees`, `puffer`, `banana`, `West-Highland-white-terrier`, `whippet`, `brown-bear`, `snowplow`, `tarantula`, `space-heater`, `sports-car`, `jean`, `sandbar`, `perfume`, `papillon`, `triceratops`, `barrow`, `peacock`, `digital-watch`, `carton`. The adversarial perturbations are normalized to $[0, 1]$ for the sake of visualization. Some of them are treated as being "visually imperceptible" based on the commonly used threshold $8/255 = 0.0314$ for $\ell_\infty$ ('linf') norms. If QuadAttacK fails using a model for a $K$ (e.g. topK=25 for DenseNet121), we leave it blank. For the benign image, the top-30 predictions by the four models respectively are:
• **ResNet50**: redshank, ruddy turnstone, red-backed sandpiper, dowitcher, oystercatcher, grey whale, red-breasted merganser, crane, sea lion, chainlink fence, lakeside, wreck, quail, partridge, screwdriver, plastic bag, pelican, parachute, killer whale, sulphur-crested cockatoo, African crocodile, white stork, pole, bucket, caldron, hummingbird, sandbar, king penguin, nail, syringe.
• **DenseNet121**: redshank, ruddy turnstone, red-backed sandpiper, oystercatcher, breakwater, dowitcher, sea lion, academic gown, abaya, mortarboard, red-breasted merganser, lifeboat, cloak, espresso, lipstick, theater curtain, wood rabbit, umbrella, refrigerator, ruffed grouse, king penguin, partridge, sandbar, diamondback, hen-of-the-woods, wine bottle, mailbox, stone wall, volcano, redbone.
• **ViT-B**: redshank, red-backed sandpiper, ruddy turnstone, dowitcher, oystercatcher, water ouzel, Madagascar cat, chain saw, apiary, red-breasted merganser, Tibetan mastiff, cicada, seat belt, American egret, wall clock, mask, snow leopard, schipperke, potter's wheel, lycaenid, mud turtle, curly-coated retriever, dumbbell, television, strainer, feather boa, buckle, junco, boa constrictor, volcano.
• **DEiT-B**: redshank, ruddy turnstone, red-backed sandpiper, dowitcher, oystercatcher, red-breasted merganser, warthog, worm fence, Indian elephant, African crocodile, maze, badger, snowplow, American black bear, stone wall, king penguin, car wheel, rock python, water ouzel, guillotine, wild boar, centipede, diamondback, apiary, barrow, horned viper, sundial, guenon, bustard, skunk.

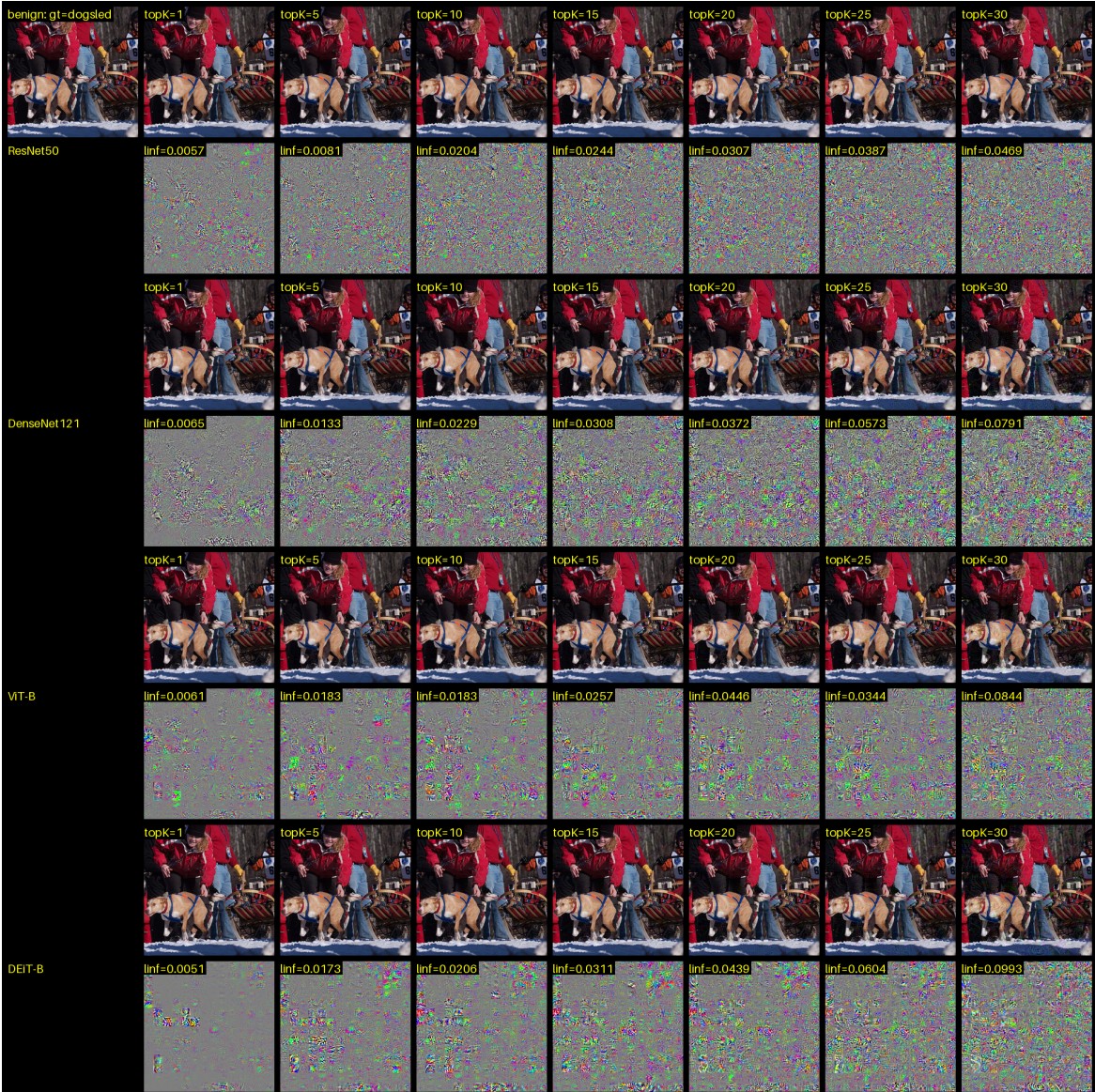

*Figure 4.* **RisingAttacK** examples of adversarial examples and associated perturbations learned for a benign image (ILSVRC2012_val_00002266 with the ground-truth label, dogsled) using a list of randomly sampled 30 targets (see the list for **seed=52** in the Appendix D) in the order of: crutch, wood-rabbit, prison, jigsaw-puzzle, joystick, space-bar, lumbermill, West-Highland-white-terrier, English-springer, spindle, hot-pot, African-hunting-dog, beaver, loggerhead, Cardigan, torch, radio-telescope, strawberry, beagle, chain, dalmatian, tiger, beer-bottle, maillot, ping-pong-ball, bighorn, hard-disc, analog-clock, hair-slide, parachute. The adversarial perturbations are normalized to $[0, 1]$ for the sake of visualization. Some of them are treated as being "visually imperceptible" based on the commonly used threshold $8/255 = 0.0314$ for $\ell_\infty$ ('linf') norms. For the benign image, the top-30 predictions by the four models respectively are:

• **ResNet50:** dogsled, Eskimo dog, bobsled, Ibizan hound, Labrador retriever, EntleBucher, beagle, Weimaraner, Greater Swiss Mountain dog, bloodhound, stretcher, Cardigan, Walker hound, redbone, Leonberg, Siberian husky, English foxhound, Chihuahua, shovel, Bernese mountain dog, malinois, ski mask, groenendael, Chesapeake Bay retriever, curly-coated retriever, drum, cocker spaniel, Gordon setter, Saluki, cowboy hat.

• **DenseNet121:** dogsled, Ibizan hound, Chesapeake Bay retriever, American Staffordshire terrier, whippet, Weimaraner, bobsled, vizsla, snowmobile, drum, malinois, Rhodesian ridgeback, Saluki, Eskimo dog, ski, Labrador retriever, mountain tent, Irish terrier, toyshop, shovel, muzzle, ski mask, dingo, alp, Irish wolfhound, Greater Swiss Mountain dog, Brittany spaniel, hog, Staffordshire bullterrier, Siberian husky.

• **ViT-B:** dogsled, Ibizan hound, Eskimo dog, American Staffordshire terrier, whippet, Greater Swiss Mountain dog, snowmobile, EntleBucher, boxer, Saluki, bobsled, Siberian husky, Norfolk terrier, Staffordshire bullterrier, basenji, Great Dane, Rhodesian ridgeback, Irish terrier, Brittany spaniel, Tibetan terrier, Chihuahua, muzzle, vizsla, beagle, rugby ball, Walker hound, Norwich terrier, Italian greyhound, Cardigan, Weimaraner.

• **DEiT-B:** dogsled, Eskimo dog, EntleBucher, Ibizan hound, whippet, Chihuahua, Weimaraner, Siberian husky, bearskin, Greater Swiss Mountain dog, Italian greyhound, bobsled, manhole cover, beagle, snowmobile, coffeepot, scabbard, bald eagle, langur, wing, espresso, stethoscope, mortarboard, dingo, suit, cowboy hat, piggy bank, carpenter's kit, basenji, zucchini.

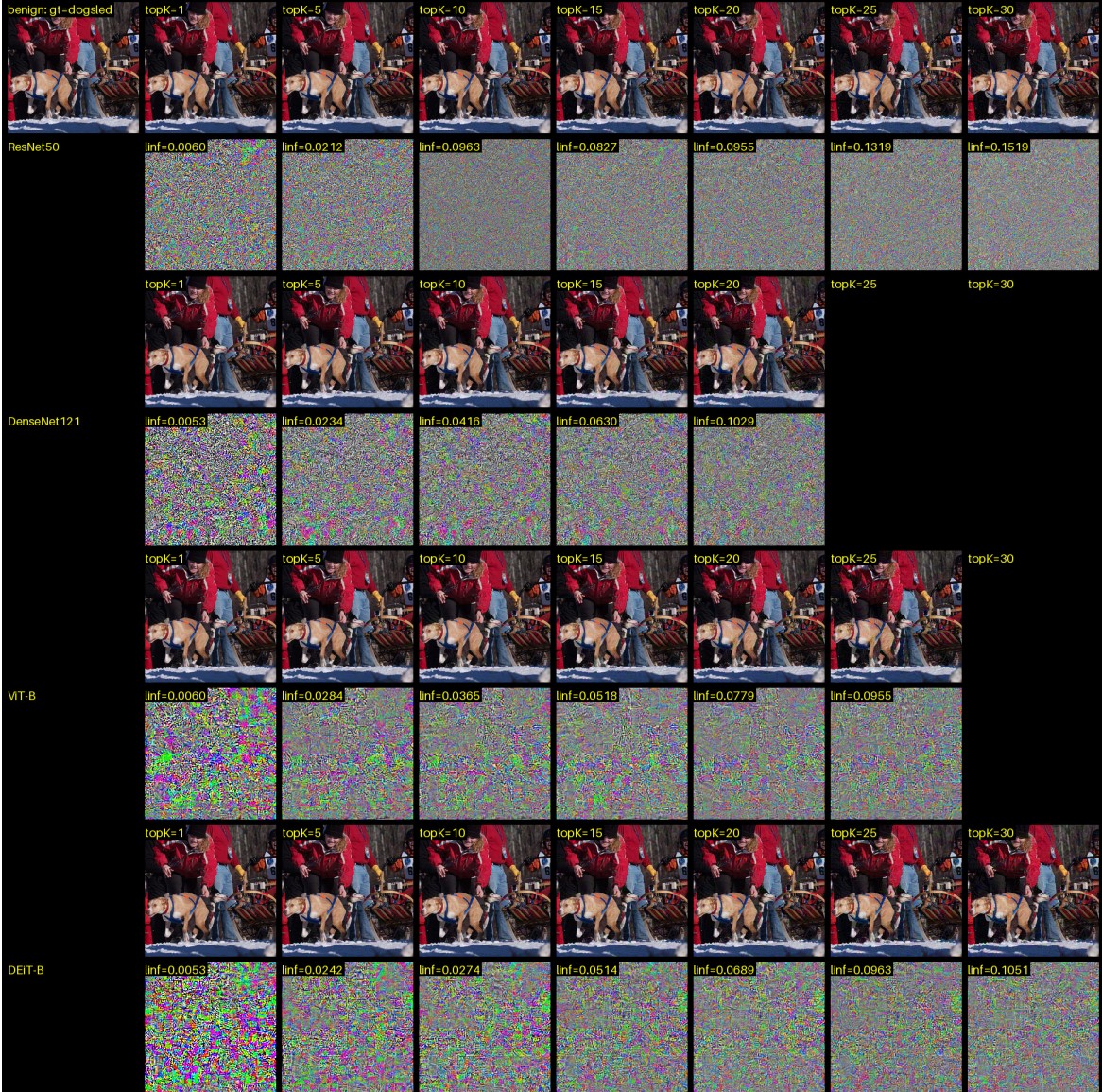

*Figure 5.* **QuadAttacK** examples of adversarial examples and associated perturbations learned for a benign image (ILSVRC2012_val_00002266 with the ground-truth label, `dogsled`) using a list of randomly sampled 30 targets (see the list for **seed=52** in the Appendix D) in the order of: `crutch, wood-rabbit, prison, jigsaw-puzzle, joystick, space-bar, lumbermill, West-Highland-white-terrier, English-springer, spindle, hot-pot, African-hunting-dog, beaver, loggerhead, Cardigan, torch, radio-telescope, strawberry, beagle, chain, dalmatian, tiger, beer-bottle, maillot, ping-pong-ball, bighorn, hard-disc, analog-clock, hair-slide, parachute.` The adversarial perturbations are normalized to $[0, 1]$ for the sake of visualization. Some of them are treated as being "visually imperceptible" based on the commonly used threshold $8/255 = 0.0314$ for $\ell_\infty$ ('linf') norms. If QuadAttacK fails using a model for a $K$ (e.g. topK=25 for DenseNet121), we leave it blank. For the benign image, the top-30 predictions by the four models respectively are:

• **ResNet50:** dogsled, Eskimo dog, bobsled, Ibizan hound, Labrador retriever, EntleBucher, beagle, Weimaraner, Greater Swiss Mountain dog, bloodhound, stretcher, Cardigan, Walker hound, redbone, Leonberg, Siberian husky, English foxhound, Chihuahua, shovel, Bernese mountain dog, malinois, ski mask, groenendael, Chesapeake Bay retriever, curly-coated retriever, drum, cocker spaniel, Gordon setter, Saluki, cowboy hat.

• **DenseNet121:** dogsled, Ibizan hound, Chesapeake Bay retriever, American Staffordshire terrier, whippet, Weimaraner, bobsled, vizsla, snowmobile, drum, malinois, Rhodesian ridgeback, Saluki, Eskimo dog, ski, Labrador retriever, mountain tent, Irish terrier, toyshop, shovel, muzzle, ski mask, dingo, alp, Irish wolfhound, Greater Swiss Mountain dog, Brittany spaniel, hog, Staffordshire bullterrier, Siberian husky.

• **ViT-B:** dogsled, Ibizan hound, Eskimo dog, American Staffordshire terrier, whippet, Greater Swiss Mountain dog, snowmobile, EntleBucher, boxer, Saluki, bobsled, Siberian husky, Norfolk terrier, Staffordshire bullterrier, basenji, Great Dane, Rhodesian ridgeback, Irish terrier, Brittany spaniel, Tibetan terrier, Chihuahua, muzzle, vizsla, beagle, rugby ball, Walker hound, Norwich terrier, Italian greyhound, Cardigan, Weimaraner.

• **DEiT-B:** dogsled, Eskimo dog, EntleBucher, Ibizan hound, whippet, Chihuahua, Weimaraner, Siberian husky, bearskin, Greater Swiss Mountain dog, Italian greyhound, bobsled, manhole cover, beagle, snowmobile, coffeepot, scabbard, bald eagle, langur, wing, espresso, stethoscope, mortarboard, dingo, suit, cowboy hat, piggy bank, carpenter's kit, basenji, zucchini.

