# OpenReview forum: "Adversarial Perturbations Are Formed by Iteratively Learning Linear Combinations of the Right Singular Vectors of the Adversarial Jacobian"
_ICML.cc/2025/Conference — ICML 2025 poster_

### Official Review · Reviewer_D2j5 · 2025-03-11

**Overall Recommendation:** 2

**Summary:**

This paper introduces RisingAttack, a novel method for generating ordered top-K adversarial attacks on Deep Neural Networks (DNNs) by optimizing directly in the image space. The method leverages Sequential Quadratic Programming (SQP) to manipulate adversarial perturbations as linear combinations of the right singular vectors of a specially constrained logit-to-image Jacobian matrix. Demonstrated across multiple architectures like ResNet-50, DenseNet-121, ViT-B, and DEiT-S on the ImageNet-1k dataset, RisingAttack outperforms existing methods such as QuadAttack by achieving higher attack success rates and less perceptible perturbations. By addressing both the ordered top-K attack problem and optimizing in the high-dimensional image space, this work enhances the understanding of adversarial vulnerabilities in DNNs.

**Claims And Evidence:**

See strengths and weaknesses.

**Essential References Not Discussed:**

See strengths and weaknesses.

**Experimental Designs Or Analyses:**

See strengths and weaknesses.

**Methods And Evaluation Criteria:**

See strengths and weaknesses.

**Other Comments Or Suggestions:**

See strengths and weaknesses.

**Other Strengths And Weaknesses:**

Strengths:
1. The approach is based on solid mathematical foundations, particularly the application of Sequential Quadratic Programming (SQP) and Singular Value Decomposition (SVD) to optimize the adversarial attack, ensuring that the perturbations are not just random but strategically effective.

2. It has been tested and shown effective across different types of deep neural network architectures, including both traditional convolutional networks and more recent transformer-based models, indicating its adaptability and broad applicability.

3. The method provides new insights into the susceptibility of DNNs to adversarial attacks, contributing to the broader understanding of model weaknesses and paving the way for developing stronger defensive mechanisms.

Weaknesses
1. The reliance on computationally intensive techniques like SQP and SVD may render RisingAttack less feasible for real-time or on-device applications where computational resources are limited. The paper does not sufficiently address the method's performance or feasibility in real-world scenarios, where computational efficiency and the ability to operate under diverse and unpredictable conditions are crucial.

2. The effectiveness of the attack largely hinges on the accurate computation of the Jacobian matrix, which can vary significantly depending on the model's architecture and the data, potentially limiting the method's effectiveness in less ideal conditions.

3. The generalization of the method to tasks outside of image classification or to non-visual data remains untested, which may limit its utility in broader applications of AI. Although it outperforms certain established methods, the paper could benefit from a broader comparison with a wider range of adversarial attack strategies to more comprehensively position its effectiveness within the field.

**Questions For Authors:**

See strengths and weaknesses.

**Relation To Broader Scientific Literature:**

See strengths and weaknesses.

**Theoretical Claims:**

See strengths and weaknesses.

---

> ### Author Rebuttal · Authors · 2025-03-31
>
> Dear Reviewer D2j5,
>
> Thank you for your valuable comments and efforts. We address your concerns as follows and will update those carefully in the revision.
>
> ### C1: The reliance on computationally intensive techniques like SQP and SVD may render RisingAttack less feasible for real-time or on-device applications where computational resources are limited. The paper does not sufficiently address the method's performance or feasibility in real-world scenarios, where computational efficiency and the ability to operate under diverse and unpredictable conditions are crucial.
>
> > Thank you. We acknowledge the importance of developing methods that can be deployed in real-world scenarios under various conditions.
>
> > Both our RisingAttacK and the prior art, QuadAttacK can not achieve real-time attacks yet (please see our time complexity report in addressing the C1 to the Reviewer ubo3).
>
> > The novelty of our proposed RisingAttacK lies in its ability of learning ordered top-K adversarial perturbations directly in the image space using SQP for the first time to our knowledge, which is crucial for enhancing our understanding of adversarial perturbations and potentially for developing adversarial defense methods.
>
> > We hope that the deployment aspects of our RisingAttacK, as well as other attack methods,  could be improved gradually when techniques become more mature with better optimizers and engineered implementations.
>
> > We also hope that the current lack of real-time running speed of our RisingAttacK (as well as the prior art, QuadAttacK) is not a factor that negatively impact us and other researchers on investigating rigorous optimization frameworks such as SQP in adversarial learning.
>
> ### C2: The effectiveness of the attack largely hinges on the accurate computation of the Jacobian matrix, which can vary significantly depending on the model's architecture and the data, potentially limiting the method's effectiveness in less ideal conditions.
>
> > Thank you. This might be a misunderstanding. The computation of the Jacobian matrix is not approximated. For most DNNs that are fully differentiable and since we are studying white-box attacks, the logits-to-input Jacobian matrix can be exactly computed. For example, we used the `torch.func.jacrev`  [1] in our implementation. So, the Jacobian computation does not limit our method's effectiveness.
>
> > The approximation component in our proposed method lies in the first-order linearization of a DNN (Eqn.6) to relax the optimization problem with highly non-linear constraints to one with linear constraints, as commonly done under the SQP framework.
>
> ### C3: The generalization of the method to tasks outside of image classification or to non-visual data remains untested, which may limit its utility in broader applications of AI. Although it outperforms certain established methods, the paper could benefit from a broader comparison with a wider range of adversarial attack strategies to more comprehensively position its effectiveness within the field.
>
> > Thank you. We acknowledge the importance of developing attack methods and testing them for many tasks, rather than image classification.  However, image classification is still the most widely used application task for studying new adversarial attack methods.
>
> > For the ordered top-K targeted attacks studied in this paper, it is still under active research with two previous approaches that we can find (Zhang & Wu, 2020; Paniagua et al., 2023). For practical considerations of conducting experimental comparisons, we follow the prior art to focus on image classification tasks. The reported experimental results are sufficiently comprehensive to verify the effectiveness of our proposed method in our opinion.
>
> > With a forward-looking thinking, our proposed method should have similar potential extensions as methods such as PGD (projected gradient descent) for learning top-1 targeted attacks. We leave those applications for further work, while we are continuing working on improving the efficiency of our current RisingAttacK.
>
>
> ---
> [1] https://pytorch.org/docs/stable/generated/torch.func.jacrev.html

---

> > ### Comment · Reviewer_D2j5 · 2025-04-08
> >
> > Thanks for the author's rebuttal. The author's response did not solve all my questions well, so I kept my previous rating.

---

### Official Review · Reviewer_69zR · 2025-03-13

**Overall Recommendation:** 3

**Summary:**

In this work, the authors propose an ordered top-K targeted white-box attack called RisingAttacK by solving the non-linearly constrained optimization problem in image space under the sequential quadratic programming framework. Experiments on ImageNet-1k dataset validate the effectiveness of RisingAttacK.

**Claims And Evidence:**

Yes

**Essential References Not Discussed:**

N/A

**Experimental Designs Or Analyses:**

Yes, the experimental design is good.

**Methods And Evaluation Criteria:**

Yes

**Other Comments Or Suggestions:**

N/A

**Other Strengths And Weaknesses:**

1. I cannot figure out why we need the top-K targeted attack. Can you provide any practical scenarios or its benefits compared with targeted/untargeted attacks?

2. It is not clear how you solve Eq. (25). Which optimizer do you adopt?

3. The proposed holistic figure of merits (FoM) metric seems to be limited. It is constrained with a single baseline. How could I calculate it with multiple baselines?

**Questions For Authors:**

N/A

**Relation To Broader Scientific Literature:**

RisingAttacK achieves good performance on top-k targeted attack compared with the baseline QuadAttack.

**Theoretical Claims:**

N/A

---

> ### Author Rebuttal · Authors · 2025-03-31
>
> Dear Reviewer 69zR,
>
> Thank you for valuable comments and efforts. We address your concerns one by one as follows. We will carefully update those in the  revision.
>
> ### C1: I cannot figure out why we need the top-K targeted attack. Can you provide any practical scenarios or its benefits compared with targeted/untargeted attacks?
>
> > First of all, targeted attacks can provide better controllability for adversaries against untargeted attacks, and are a harder problem to solve, as widely recognized in the literature.
>
> > For targeted attacks, in terms of how precisely and how aggressively we could manipulate the outputs (i.e., the entire logits) of DNNs, there are three different settings with increasing difficulties to achieve: (i) conventional top-1 targeted attacks, (ii) unordered top-$K$ targeted attacks ($K\geq 1$), and (iii) ordered top-$K$ targeted attacks ($K\geq 1$). We focus on (iii) in this paper.
>
> > From top-1 to top-$K$ (unorder or ordered), as pointed out in (Zhang & Wu, 2020), the robustness of attack methods themselves  should be investigated. For example, a top-1 targeted attack may be viewed successfully (e.g., from a cat image to a dog prediction), the ground-truth label (cat) may still be the top-2 or 3 prediction, which will be less effective in terms of top-k (e.g., top-5) accuracy metric. So, top-$K$ attacks, especially with a large $K$, can ensure the ground-truth labels will be pushed sufficiently away.
>
> > Between unordered top-$K$ and ordered top-$K$, as pointed out in (Paniagua et al., 2023), there are two scenarios in practice, ordinal examples and nominal examples:
> >> ``Imagine a cancer risk assessment tool that analyzes 2D medical images (e.g., mammograms) to categorize patients’ cancer risk into the ordinal 7-level risk ratings ([Extremely High Risk, Very High Risk, High Risk, Moderate Risk, Low Risk, Minimal Risk, No Risk]), An oncologist could use this tool to triage patients, prioritizing those in the highest risk categories for immediate intervention. An attacker aiming to delay treatment might use an ordered top-3 adversarial attack to change a prediction for a patient initially assessed as Very High Risk. They could target the classes [Moderate, Low, Minimal], subtly downgrading the urgency without breaking the logical sequence of risk categories. An unordered attack, in contrast, might lead to a sequence like [Low, Very High, Minimal], disrupting the ordinal relationship between classes. Such a disruption could raise red flags, making the attack easier to detect."
>
> >> Please see page 2 for the nominal examples in (Paniagua et al., 2023), due to space limit, we can not quote those examples here.
>
> > Similar in spirit to the ordinal and nominal examples provided in (Paniagua et al., 2023), learning ordered top-$K$ targeted attacks could find practical applications such as for the recommendation systems  (that often recommend a number of items in a particular order) and the retrieval systems (that often return ordered retrieved items).
>
> > More related to computer vision applications,  APIs such as Google Cloud Vision, Microsoft Azure Computer Vision, Amazon Rekognition, and IBM Watson Visual Recognition, often return ordered top-$K$ (e.g., 10) predictions about input images, for which  ordered top-K targeted attacks could be studied.
>
> ### C2: It is not clear how you solve Eq. (25). Which optimizer do you adopt?
>
> > Thank you. As stated in lines 340-341: ``We show that Eqn. 25 has a closed-form solution (see the proof in Appendix C), reproducing the result in Eqn. 18.". So, we do not need an optimizer to solve it.
>
> ### C3: The proposed holistic figure of merits (FoM) metric seems to be limited. It is constrained with a single baseline. How could I calculate it with multiple baselines?
>
> > Thank you. The propose FoM is for pair-wise comparisons. For comparing a method 1 against other methods (2 to M), there are three possible extensions:
> + We may simply compute $mean_{j=2}^M FoM(1, j)$, similar in spirit to the mean Average Precision (mAP) that is widely used in object detection for comparing accuracy across multiple categories and across multiple IoU thresholds.
> + We may further consider: (i) the strict $\text{FoM} = \frac{\text{ASR}^1}{\max_{j=2}^M(\text{ASR}^j)}\cdot \frac{1}{3}\cdot \sum_{p\in \{1, 2,\infty\}}\frac{\min_{j=2}^M\ell_p^j}{\ell_p^1}$, to show how well the method 1 performs against the best of the rest in terms of every aspects (ASR and three norms), and (ii) the  average $\text{FoM} = \frac{\text{ASR}^1}{mean_{j=2}^M(\text{ASR}^j)}\cdot \frac{1}{3}\cdot \sum_{p\in \{1, 2,\infty\}}\frac{mean_{j=2}^M\ell_p^j}{\ell_p^1}$

---

### Official Review · Reviewer_ubo3 · 2025-03-13

**Overall Recommendation:** 4

**Summary:**

The paper introduces a new method for generating ordered top-K adversarial attacks. The authors use Sequential Quadratic Programming (SQP) to solve the optimization problem behind top-K adversarial attacks directly in the image space. After adapting the SQP algorithm to make the computation tractable and avoid high $\ell_infty$ norm, they derived a method called RisingAttacK that outperforms the previously state-of-the-art method QuadAttacK on ImageNet-1k. The new adversarial method provides insights on the nature of the adversarial attacks, they are linear combinations of the right singular vectors of the attack-targets-ranking constrained logit-to-image Jacobian matrix.

## update after rebuttal

The authors answered the questions that I had about the two claims that I identified in the paper. Concerning C1, the authors did some preliminary experiments to address my question and are willing to "re-run all the experiments with time complexity recorded for a full-scale time complexity comparison between the two methods in revision". I am satisfied with the answer to my question about C1. Concerning C2, the authors agreed with my remark that the claim is misleading and suggested to change the title of the paper as a consequence. The rephrasing of the claim makes it less strong, the claim "Adversarial Perturbations Are Linear Combinations of the Right Singular Vectors of the Attack-Targets-Ranking Constrained Jacobian" was an interesting theoretical claim about adversarial attacks. While I appreciate that the authors took into account my remark, the fact that the original claim was misleading and will be rephrased impact the message of the paper.

Overall, I am satisfied with the authors' response and have decided to change my score from 3 to 4.

**Claims And Evidence:**

There are two main claims in the paper:
1.	"(…) ordered top-K adversarial perturbations can be expressed as linear combinations of the right singular vectors (corresponding to non-zero singular values) of the attack-targets-ranking constrained logit-to-image Jacobian matrix."
2.	"Our RisingAttacK significantly outperforms the previous state-of-the-art approach, QuadAttacK, consistently across all top-K (1, 5, 10, 20, 25) and four models (ResNet-50, DenseNet-121, ViT-B and DEiT-S) on ImageNet-1k in experiments."

The first claims provides a valuable insight on the structure of a top-K adversarial attack by defining the low dimensional ($O(d)$) manifold it belongs to. However, given the iterative nature of the method, the adversarial perturbation after reaching the iterations budget is actually a linear combination of the right singular vectors of A (attack-targets-ranking constrained logit-to-image Jacobian matrix) evaluated on the adversarial image generated by the previous iteration. This make the subspace to which the adversarial attack belongs to hard to interpret as it depends on the adversarial attack obtained in the previous iteration.

The second claim is clearly supported by extensive experiments on different iteration budgets and datasets.

**Essential References Not Discussed:**

na

**Experimental Designs Or Analyses:**

See "methods and evaluation criteria"

**Methods And Evaluation Criteria:**

The choice of architectures and dataset makes sense. The FoM metric gives a nice overview on how methods compare in different scenarios. The authors say that they compare attacks under the same computing budget for fair comparison. However this computation budget is measured by the number of iterations performed by the method. To ensure that the number of iterations is a good proxy for computation budget, it would have been valuable to provide the time complexity of one iteration for each method or eventually an empirical measurement of the compute time and memory required.

**Other Comments Or Suggestions:**

na

**Other Strengths And Weaknesses:**

na

**Questions For Authors:**

•	Would it be possible to report the time complexity of an iteration of RisingAttacK and the time complexity of an iteration of QuadAttacK ? Or alternatively, evaluate empirically the time necessary to run both methods ? This is supported by having fair comparison between methods (see "Methods And Evaluation Criteria").
•	Could you clarify the claim of the paper "(…) ordered top-K adversarial perturbations can be expressed as linear combinations of the right singular vectors (corresponding to non-zero singular values) of the attack-targets-ranking constrained logit-to-image Jacobian matrix." ? The iterative nature of the method seems to make the interpretation of this right singular vectors more difficult (see "Claims and Evidences" for more details)

**Relation To Broader Scientific Literature:**

This work is part of the larger family of work on adversarial attacks. Earlier work mainly focused on perturbing the top-1 prediction on a neural network. In this work the authors studied the case ordered top-K attack. Top-K attacks have been introduced by Zhang and Whu [1], Paniagua et al.[2] improved the attacks method using quadratic programming. The authors build upon the method of Paniagua et al.[2].

[1] Zhang, Zekun, and Tianfu Wu. "Learning ordered top-k adversarial attacks via adversarial distillation." Proceedings of the IEEE/CVF conference on computer vision and pattern recognition workshops. 2020.
[2] Paniagua, Thomas, Ryan Grainger, and Tianfu Wu. "QuadAttac $ K $: A Quadratic Programming Approach to Learning Ordered Top-$ K $ Adversarial Attacks." Advances in Neural Information Processing Systems 36 (2023): 48962-48993.

**Theoretical Claims:**

na

---

> ### Author Rebuttal · Authors · 2025-03-31
>
> Dear Reviewer ubo3,
>
> Thank you very much for your valuable comments and efforts. We address your concerns one by one as follows, which will be carefully updated in revision.
>
> ### C1: Report the time complexity of an iteration of RisingAttacK and that of QuadAttacK
>
> > Thank you. We report the **average seconds/iteration** using a batch of 32 images over 30 iterations on a single A100 GPU as follows,
> | $K$ | QuadAttacK | RisingAttacK |
> | :----:| :----:|:----:|
> | 1 | 1.31 | 1.38 |
> | 5 | 1.17 | 0.83 |
> | 10 | 0.95 | 1.57 |
> | 20 | 1.30 | 2.8|
> | Avg| 1.18 | 1.64 |
>
> > We note three aspects,
>
> + For both methods, the time complexity of an iteration is not necessarily monotonically related to $K$. It reflects the complexity of the underlying QP to be solved, which in turn is affected by the the sheer challenges of the randomly sampled top-$K$ attack targets for different $K$'s and different images.
> + On the average (32 images over 30 iterations), our RisingAttacK is more computationally expensive than QuadAttacK. With the significantly improved performance by our RisingAttacK, the increased time complexity seems to be reasonable.
> + We will re-run all the experiments with time complexity recorded for a full-scale time complexity comparison between the two methods in revision.
>
> ### C2: Could you clarify the claim of the paper "(…) ordered top-K adversarial perturbations can be expressed as linear combinations of the right singular vectors (corresponding to non-zero singular values) of the attack-targets-ranking constrained logit-to-image Jacobian matrix." ? The iterative nature of the method seems to make the interpretation of this right singular vectors more difficult.
>
> > Thank you for pointing this out. We agree with you that the iterative nature of our RisingAttacK indeed makes the interpretation not precise.
>
> > We propose to change the title to ``*Adversarial Perturbations Are **Formed by Iteratively Learning** Linear Combinations of the Right Singular Vectors of the Attack-Targets-Ranking Constrained Jacobian*", as well as all the related statements in text.
>
> >> We would like to continue to highlight the observation of "*Linear Combinations of the Right Singular Vectors of the Attack-Targets-Ranking Constrained Jacobian*" in solving Eqn.18 at each iteration, which we think provides useful insights for understanding adversarial perturbations directly in the image space.

---

### Decision · Program_Chairs · 2025-05-01

**Decision:**

Accept (poster)

**Comment:**

One reviewer brought up several claims were resolved during the discussion period. Unfortunately, the other two reviewers barely engaged in the discussion. Upon closer inspection, I believe their concerns to be either answered, unaddressable hypothesizing, or valid concerns of relatively minor importance. As a result, I am inclined to be aligned with the positive reviewer in this case, who gave a well thought-out critique of merits of the paper, and is now happy to accept the paper, based on the authors promise to update the wording around their claims to be less misleading and to finalize the full scale of the additional experiment.